# VeriBench: End-to-End Formal Verification Benchmark for AI Code Generation in Lean 4

## Abstract

Formal code verification offers a path to provably correct software, but evaluating language models' capabilities in this domain requires comprehensive benchmarks. Provably correct code would eliminate entire classes of vulnerabilities, mitigate critical system failures, and potentially transform software engineering practices through inherently trustworthy implementation methodologies. We present VeriBench, a benchmark for assessing **end-to-end** formal code verification in Lean 4, requiring models to generate complete program implementations with tests, specifications/theorems, and machine-checked proofs from Python references. Our benchmark comprises 140 tasks across five difficulty levels: 56 HumanEval problems, 41 foundational programming exercises, 10 classical algorithms, 28 security-critical programs adapted from real-world vulnerabilities and 5 programs from the Python standard library. To enable comprehensive capability assessment, we establish four hierarchical evaluation subtasks with explicit metrics: (1) Lean 4 compilation success, (2) proportion of unit test passing, (3) correctness-theorem synthesis quality, and (4) proof success rates with pass@1. However, evaluation reveals significant limitations in current models: Claude 3.7 Sonnet achieves only 35.0% compilation success but 40.6% of unit test passing – while LLaMA-70B fails to compile any programs despite 50 feedback-guided attempts on a previous version of VeriBench. Models demonstrate similar performance on theorem evaluation, reaching 0.615% theorem accuracy as measured by a LLM judge. However, proof generation remains particularly challenging—our DSP (Draft Sketch Proof) proving agent achieves only 28.9% pass@1. In contrast, our trace-based self-debug agent architecture achieves 49.3% compilation success, demonstrating the potential of iterative, feedback-driven approaches. To enable scalable evaluation, we introduce a novel methodology for certifying the trustworthiness of LLM judges. We validate our theorem/specification LLM judge by applying a novel trustworthiness methodology that verifies adherence to fundamental logical properties, such as consistency and monotonicity, thereby facilitating reliable, automated generation of theorems and specifications. VeriBench establishes a rigorous foundation for developing AI systems capable of synthesizing provably correct, bug-free code, thereby advancing the trajectory toward more secure and dependable software infrastructure.

## 1 Introduction

Large language models (LLMs) have demonstrated impressive capabilities in code generation, leading to widespread integration into developer workflows. As a result, AI-generated code is making its way into commercial software systems and may soon occupy a large portion of publicly existing code. Despite their usefulness, LLMs are inherently probabilistic and cannot guarantee the correctness of the code they produce. Consequently, such code often contains bugs, ranging from logical flaws to serious security vulnerabilities (Perry et al., 2023; Team, 2025; Claburn; Pearce et al., 2022). As adoption grows, these errors risk becoming a major obstacle to developer productivity, since human review is typically needed to identify and fix them (Tambon et al., 2024; Nguyen et al., 2022; Srivatsa et al., 2024; GitHub Engineering, 2023).

Formal verification – the process of mathematically proving that an implementation adheres to a precise specification – offers a principled path to automatic verification and trustworthy AI. Historically,

it has been applied mainly in high-assurance settings such as hardware design (processor verification) (Reid, 2016), aerospace avionics (Holzmann & James, 2006), medical-device firmware (Alur et al., 2010), nuclear-power control systems (Linnosmaa et al., 2023), and mission-critical financial or smart-contract infrastructure (Team, 2023), where the cost of failure far outweighs the cost of formal analysis. A *formal specification* is a mathematical statement of the intended behavior of a program (e.g., `reverse` returns a list whose length equals the length of the input list), and a *machine-checkable proof* is a logical argument that a proof assistant's kernel can verify automatically. Unlike general theorem proving in mathematics, code verification focuses on proving properties about computational processes and their implementations.

LLMs have the potential to democratize formal verification by co-generating implementations, their specifications/theorems, and the proofs that connect them, bringing strong correctness guarantees to everyday software development.

To unlock the full potential of verifiable code generation, the development of robust benchmarks is essential to measure progress and guide future research. However, designing such benchmarks is challenging: verifiable code generation encompasses several interdependent components, including the synthesis of code, formal specifications, and accompanying proofs. High-quality data must be curated for each of these components, along with rigorous, task-specific evaluation metrics. Therefore, we use Trace agents (Cheng et al., 2024) to bootstrap the generation of aligned Python functions with docstring specifications and corresponding Lean proofs.

Our goal with **VeriBench** is to close two gaps left open by prior work. First, existing verification datasets—such as VERINA, FVAPPS, CLEVER, DafnyBench, and CloverBench (Chen et al., 2021; Ye et al., 2025; Dougherty & Mehta, 2025; Thakur et al., 2025; Loughridge et al., 2024; Sun et al., 2024)—are populated almost exclusively with textbook algorithms or synthetic exercises. VeriBench is the benchmark *first* to include security-critical programs written by developers: its *SecuritySet* adapts buffer-overflow, privilege-escalation, and race-condition labs from MIT 6.858 (CSAIL, 2024), so models must eliminate real vulnerabilities instead of toy bugs and aim to simulate real code that people want verified in practice. In addition we also feature a *RealCodeSet* that features Python standard library code that enables evaluation on real code used in production. We discover model cannot prove a single theorem in this set. Second, earlier suites report only single-shot or best-of-$k$ scores; they never benchmark an explicit loop that reads verifier feedback, rewrites the artifact, and resubmits until the compilation succeeds. VeriBench, therefore, is the first to illustrate agentic evaluation with a reference Trace-based (Cheng et al., 2024) framework (baseline, self-debug, self-improve) built on generative optimization traces, making feedback-driven, closed-loop verification a primary evaluation.

Besides the *SecuritySet* and *RealCodeSet*, VeriBench adds *EasySet* and *CSSet*, which target foundational reasoning and classical algorithms, respectively, giving the benchmark a diverse topic mix and a broad difficulty range (see §3). We focus on translating *existing* Python code and its docstrings to Lean because millions of such snippets already live in open-source repos; turning this real-world corpus into machine-checked programs fixes latent bugs directly, whereas starting from a fresh natural-language description would add an unnecessary detour that today's LLMs no longer require.

Unlike the static-analysis, linting, and fuzz-testing pipelines already common in industry, Lean 4 supplies compile-time, machine-checked proofs that hold for *all* inputs and compile into the shipped binary, providing end-to-end correctness guarantees that conventional tools cannot match.

Our **contributions** are:

1. **Production-grade code validation.** The benchmark includes 5 programs taken from the Python standard library, enabling evaluation on real code used in production.

2. **A security-grounded benchmark.** VeriBench is the first Lean 4 dataset to include developer-written, security-critical programs (buffer overflow, privilege escalation, and race condition labs from MIT 6.858), complementing textbook and synthetic tasks.

3. **Diverse task spectrum.** Five subsets — *Easy Set*, *CS Set*, *Real Code*, *HumanEval Set* and *Security Set* — cover everything from basic reasoning to classical algorithms, real-world exploits and even real code used in production – enabling fine-grained difficulty analysis.

4. **End-to-End agent evaluation.** We provide a DSPy-React agent and Trace-based reference agent (baseline, self-debug, self-improve) that interacts with the Lean compiler and judge, turning feedback-driven verification into a measurable axis of performance.

5. **Comprehensive Lean artifacts.** Each problem ships runnable Python code, a gold Lean implementation, unit tests, correctness theorems, and machine-checked proofs—yielding the first end-to-end yardstick for provably correct code generation.

## 2 RELATED WORK

**Benchmarks for Code Verification.** Loughridge et al. (2024) introduce DAFNYBENCH, the first large-scale benchmark for evaluating LLMs in formal software verification. It consists of over 750 Dafny programs (approximately 53K lines of code) stripped of verification "hints," requiring models to regenerate the missing annotations to pass the verifier. Evaluated on GPT-4, GPT-4 Turbo, Claude 3, and others, the best-performing system achieved a success rate of roughly 68%, demonstrating the potential of machine-assisted verification while highlighting performance variability with respect to program size, hint complexity, and retry strategies.

To more comprehensively evaluate the different stages of formal development, Ye et al. (2025) introduces VERINA, a 189-task Lean benchmark that jointly assesses LLM-generated code, formal specifications, and machine-checked proofs. While LLMs perform reasonably well on code and spec generation, they continue to struggle with constructing formal proofs, underscoring the need for more targeted training and architectural improvements. Addressing this need, Cao et al. (2025) proposes a decomposition of the informal-to-formal verification pipeline into six subtasks, and releases a dataset of 18K paired examples across five formal specification languages.

In parallel, Dougherty & Mehta (2025) introduce FVAPPS, a machine-generated Lean 4 benchmark of 4,715 problems stratified by assurance level. Built via a test-driven LLM pipeline, it shows that top models like Claude Sonnet and Gemini 1.5 Pro prove only 30% and 18.5% of theorems, respectively, with human-written solutions still falling short at scale. CLEVER (Thakur et al., 2025) offers a more focused challenge: 161 Lean problems requiring both a formal spec and a correctness-proofed implementation. It's non-computable, spec-agnostic setup avoids test leakage and demands true reasoning—so far, models fully solve only 1 of 161 tasks. Finally, broadening the scope beyond formal proof, Ouyang et al. (2025) introduces KERNELBENCH, a benchmark for generating optimized GPU kernels for 250 PyTorch workloads. Using profiler feedback and examples, iterative loops boost success from 12% to over 70%, showcasing the impact of reinforcement-style correction.

VeriBench distinguishes itself from existing benchmarks by targeting the full pipeline of formal code verification grounded in realistic programming tasks. Unlike FVAPPS, which consists of thousands of machine-generated Lean problems optimized for scale and raw proof success rates, VeriBench uses human-curated Python functions drawn from foundational algorithms and practical programming contexts. Each example is paired with a complete Lean 4 formalization—including functional and imperative implementations, unit tests, correctness theorems, and machine-checkable proofs—emphasizing artifact completeness over sheer volume. Compared to VERINA, which decomposes the verification process into separate subtasks like spec generation and proof synthesis, VeriBench evaluates holistic translation performance: how well models can go from informal code and natural language to executable and provable formal artifacts. Moreover, VeriBench supports the evaluation of agentic systems that iteratively refine their outputs through feedback.

## 3 VERIBENCH

**Overview.** VeriBench is a benchmark designed to evaluate the end-to-end code verification capabilities of large language models, requiring them to generate complete Lean 4 artifacts from reference Python programs or their accompanying docstrings. It encompasses a broad range of translation components, including function implementations, natural language descriptions, unit tests, theorems, formal proofs, and example input-output behavior.

Instead of relying on a deep embedding of source language semantics, VeriBench adopts a shallow embedding approach, aiming to produce formal representations that faithfully capture the behavior and intent of the original code. By offering a structured and challenging testbed, VeriBench enables

rigorous evaluation of model performance in both automated code translation and formal verification, with a particular focus on the correctness, completeness, and provability of the generated Lean 4 artifacts.

Concretely, VeriBench consists of four subsets:

1. **HumanEval** – 56 standard programming puzzles from Chen et al. (2021);

2. **EasySet** – 41 bite-size logic and intro-programming tasks;

3. **CSSet** – 10 classical data-structure and algorithm problems;

4. **SecuritySet** – 28 examples of buffer overflow, privilege escalation, and race condition labs drawn from real code.

5. **RealCodeSet** – 5 programs from the Python standard library, used to evaluate model performance on production-grade code.

This ensures coverage of tasks that range from simple correctness theorems to more complex invariants and algorithmic properties to real code people want verified in practice.

**Task.** The task is to translate Python code with its docstring and unit tests to a parallel Lean 4 implementation that compiles, but with an additional set of comprehensive correctness theorems.

## 4   GOLD STANDARD FOR LEAN 4 BENCHMARK FILES

**Purpose.** Each gold file (i) specifies the intended behavior of a Lean implementation, (ii) includes functional and imperative implementations, (iii) declares a *Pre-condition* as a predicate (without proof at declaration), (iv) states algebraic or semantic properties as separate proven theorems, (v) defines a *Post-condition* as the conjunction of these properties, (vi) proves a *Correctness* theorem ($\rightarrow$) by combining property theorems, and (vii) when an imperative version is included, proves an *Equivalence* theorem between the two implementations. This structured approach facilitates partial-credit evaluation and allows robust comparison across different theorem decompositions.

**Required order.** The gold file must first provide the pure/functional implementation $Prog$ followed by comprehensive unit tests covering both positive cases (valid input-output pairs respecting *Pre* and all properties) and negative cases (inputs or conditions violating any predicate or safety constraints). A predicate : $Input \rightarrow$ defining admissible inputs is specified next (trivially True if types enforce constraints). An exhaustive set of property theorems (identities, commutativity, safety, etc.) are then individually stated and proved. Subsequently, a *Post-condition* combining these properties is defined as $Post(x) := P_1(x) \land P_2(x) \land \cdots \land P_n(x)$, simplifying to a single property in basic cases. The *Correctness* theorem bundles all properties into one overarching statement: $\forall x \in X, ; (x) \implies Post(x, Prog)$, with a straightforward proof using `And.intro`. Finally, if applicable, an imperative implementation is provided alongside quick tests, and an *Equivalence* theorem confirms its functional equivalence with the pure version.

### 4.1   CONSTRUCTION

**VeriBench-*HumanEval*.** This subset extends the original HumanEval dataset (Chen et al., 2021), a widely used benchmark for evaluating the coding capabilities of language models. VeriBench-HumanEval transforms each Python problem into a Lean 4 formal verification task. The pipeline begins by parsing each HumanEval problem to extract the function signature, docstring, canonical implementation, and unit tests. These elements are then assembled into a clean, standalone Python file, with additional unit tests added where appropriate to cover important edge cases.

Subsequently, the same problem is translated into Lean 4 using a shallow embedding approach. Each Lean file includes (1) the Lean 4 implementation of the function, (2) a natural language docstring, (3) equivalent unit tests expressed as both `#eval` expressions and `example` theorems, and (4) one or more formal theorems that specify correctness properties.

Notably, where applicable, an imperative version of the function is also implemented in Lean 4, along with a theorem asserting its equivalence to the functional version. This structured Python–Lean 4 pairing enables automatic metric-based evaluation, including compilation success, proof validation, and

functional correctness through Lean's evaluation mechanism—thus providing a rigorous framework for assessing the ability of language models to translate, reason about, and formally verify programs.

```python
# Implementation
def my_max(a: int, b: int) -> int:
    """
    Return the larger of two non-negative integers.
    """
    return b if a <= b else a

# Tests
from typing import Callable

def check(candidate: Callable[[int, int], int]) -> bool:
    assert candidate(7, 3) == 7, f"expected 7 from (7,3) but got {candidate(7, 3)}"
    ... (other tests) ...
    return True

if __name__ == "__main__":
    assert check(my_max), f"Failed: {__file__}"
    print(f'All tests passed: {__file__}!')
```

Listing 1: An exemplar input Python code of VeriBench-EasySet (simplified for showcase).

```
namespace MyMax

-- Implementation
def myMax (a b : Nat) : Nat :=
 if _ : a ≤ b then b else a

infixl:70 "⊔"=> myMax -- left-associative, precedence 70

-- Tests
#eval myMax 7 3 -- expect 7
example : myMax 7 3 = 7 := by native_decide
#eval myMax 0 0 -- expect 0
example : myMax 0 0 = 0 := by native_decide

-- Theorems
@[simp] theorem max_left_identity (n : Nat) : myMax 0 n = n := sorry

-- Theorem Right Identity
@[simp] theorem max_right_identity (n : Nat) : myMax n 0 = n := sorry

... (other theorems) ...

end MyMax
```

Listing 2: An exemplar golden output Lean 4 code of VeriBench-EasySet.

**VeriBench-*EasySet*.** This subset provides a simplified alternative to VeriBench-HumanEval, targeting foundational programming and reasoning skills. It features a collection of clear, self-contained problems modeled after classic introductory programming exercises. Examples include computing the factorial of a number, reversing a list, checking for palindromes, finding the maximum value in a list, and counting character frequencies in a string. Each task is framed as a concise coding prompt, similar to a short exam question, and is accompanied by correct implementations in both Python and Lean 4. The Lean 4 solutions include the function definition, unit tests written using `#eval` and `example` theorems, and, when appropriate, formal theorems with accompanying proof sketches. The problems are selected to be easily checkable yet demand careful formalization, making VeriBench-EasySet a suitable benchmark for evaluating the basic formal reasoning capabilities of language models in a controlled and interpretable environment. For an example of the easy set, see appendix K

**VeriBench-*CSSet*.** This subset comprises fundamental computer science data structures and algorithms, covering essential computational problems including sorting, searching, dynamic programming, and string manipulation. The problems are drawn from core undergraduate computer science curricula, focusing on classical algorithms with well-established correctness properties and complexity characteristics. For foundational problems such as sorting, we include multiple algorithmic approaches with varying implementation complexity and runtime characteristics, ranging from simple quadratic algorithms like bubble sort and insertion sort to more sophisticated divide-and-conquer approaches like merge sort and quicksort. While these algorithms might be easy for LLMs to implement, it is very hard to formally verify that they are correct.

Each task follows the same structured format as other subsets, providing both Python reference implementations and corresponding Lean 4 translations with functional definitions, unit tests, and formal correctness theorems. The theorems capture essential algorithmic properties such as sortedness invariants, search completeness, and dynamic programming optimality conditions. For an example of the CS set, see appendix K

**VeriBench-*SecuritySet*.** To ensure VeriBench reflects the challenges of real-world, security-critical systems, we include formal translations of programs from MIT's 6.858 lab. For example, one challenge in VeriBench tests the models capabilities to translate a Python program with a buffer overflow shown below, to a Lean program without it. By incorporating these authentic examples, our benchmark emphasizes the high-assurance verification tasks that practitioners care about most. For an example of the Security set appendix K

**VeriBench-*RealCodeSet*.** To evaluate model performance on production-grade code, VERIBENCH includes a subset of functions taken verbatim from the Python standard library. Specifically, this subset features key algorithms from widely-used modules such as `bisect.py` and `heapq.py`. For instance, one challenge requires translating the `heappush` function from Python's `heapq` module into a formally verified Lean 4 implementation, complete with proofs of its heap-invariance properties. By incorporating code that is actively used in production by millions of developers, this subset tests a model's ability to reason about and formally verify implementations that are optimized for performance and correctness in real-world scenarios. For an example of the *RealCodeSet*, see appendix K.

**Curation.** We attempt to write a comprehensive set of theorem properties for each benchmark test. Since in practice this is provably impossible to guarantee (see E), to make sure we generate as many important theorems for the benchmark, we have a two stage generation pipeline were a second human curator assisted with AI makes sure the theorems are as exhaustive as possible.

## 5 EVALUATION

### 5.1 SETUP

**DSPy React Agent.** We build a simple tool-use agent with only a single DSPy (Cheng et al., 2024; Khattab et al., 2022) react module with a maximum of 50 Lean 4 tool calls. The tool given is the Lean 4 RL environment accessed via PyPantograph (Aniva et al., 2025). Any feedback from the environment (e.g., errors, code lines, etc.) is fed back to the agent to produce an output that compiles. We choose 50 to provide a model with enough budget for the task, as language models cannot generate Lean 4 accurately. As will be shown in later sections, even with such a high number of calls with feedback, all open source models obtained 0% compilation accuracy.

**Trace.** Trace lets an LLM "rewrite the agent as it learns": after each run it passes the optimizer (i) the agent's current source, (ii) the full execution trace, (iii) any verifier error messages (or an RL environment), and (iv) the resulting reward. The LLM edits the code in-context using this quartet of signals, then the refined agent runs again, closing a tight loop of self-improvement.

**LLM Trace Agent.** We use Trace (Cheng et al., 2024) to build a simple LLM agent. We provide three variants: a *baseline agent* that only outputs the theorem translation from the code. Then, we build a *self-debug agent* that adds a self-correction loop where the compilation error is provided as feedback to the agent and the agent is asked to produce correct theorems conditioned on the past incorrect result, along with debugging information to fix the mistake. We build a detailed debugging report similar to what a human would get from an IDE, with specific line number information, code snippets surrounding the line that triggered the error, and the actual error from the compiler. Finally, we built a *self-improving agent* (self-debug + judge) that has a layered design. This agent first generates a theorem translation. If there is a bug, it goes through the self-correction loop similar to the self-debug agent. If the translated theorem compiles, it will use the LLM Judge described below to get a score, and conditioned on the generated theorem and score, the agent is asked to self-improve in context to propose a better theorem to maximize the score. **LLM Judge.** The LLM judge scores the quality candidate Lean 4 translated from the Trace agent using the same LLM Trace uses (in this case Claude). The LLM judge is provided: (1) the original Python and docstring, (2) the translated Lean 4 program, (3) the translated Lean 4 unit tests, and (4) a set of candidate theorems based on 1-3.

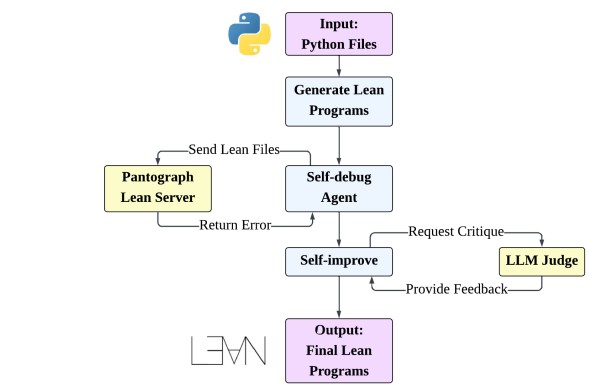

Figure 1: Our Trace-based Evaluation Framework.

The LLM judge is asked to judge if the theorems are correct and comprehensive, and if they are not it penalizes the output of the agent. Since this is done during evaluation the LLM judge does not have the gold reference Lean file and instead only has an example file in context demonstrating a good Lean 4 translation for addition. The model is prompted to score based on the Lean 4 code quality and no further computation besides the judge's autoregressive generation employed. A candidate theorem must (i) be correct – equivalent to the Python reference and docstring – and (ii) comprehensive, covering every possible property the Python and docstring imply. Note, however, that comprehensive is difficult if not impossible to guarantee even during the gold reference generation of the benchmark.

## 6 THEOREM PROVING WITH VERIBENCH

| Model + Prompting | Easy | CS | Real | HE | Security | Pass@1 |
|---|---|---|---|---|---|---|
| Claude-3.5 Sonnet v1 (2024-06-20) + DSP | 31/278 | 0/68 | 0/12 | 14/387 | 0/112 | 45/857 **5.25%** |
| Claude-3.7 Sonnet (2025-02-19) + DSP | 81/278 | 2/68 | 0/12 | 67/387 | 6/112 | 156/857 **18.2%** |
| DeepSeek-ProverV1.5-SFT (Xin et al., 2024) + prompting | 59/278 | 2/68 | 0/12 | 57/387 | 15/112 | 133/857 **15.55%** |
| DeepSeek-ProverV2-7B (Ren et al., 2025) + prompting | 114/278 | 6/68 | 0/12 | 85/387 | 43/112 | 248/857 **28.94%** |
| Goedel-Prover V2-8B (Lin et al., 2025)+ prompting | 109/278 | 9/68 | 0/12 | 76/387 | 31/112 | 225/857 **26.25%** |

Table 1: Performance of different models on VERIBENCH theorem-proving tasks. The table shows results for LLMs evaluated under the Draft, Sketch, and Prove (DSP) protocol (+ DSP) (Jiang et al., 2022) and dedicated provers evaluated using a standard single prompt (+ prompting). All results are reported at pass@1. VERIBENCH splits : Easy Set (Easy), CS Set (CS), Real Python Code (Real), HumanEval Set (HE), Security Set (Security).

Table 1 presents the results of evaluating several LLMs on theorem-proving tasks, demonstrating that VERIBENCH provides a meaningful benchmark for assessing the theorem-proving capabilities of LLMs on code verification. The evaluation dataset was constructed from the gold Lean implementation files in VERIBENCH. We performed preprocessing on these files to prepare the data for theorem-proving tasks. This preprocessing separates theorems contained in the same file and removes comments and examples to reduce context size. For each theorem, we retain all definitions and variables declared prior to the theorem, removing the remaining content.

We observe from Table 1 that the examples across the different splits remain challenging. As expected, the Easy set achieves the highest success rates, since it contains comparatively simpler problems. None of the models was able to prove theorems on real code problems. Among the evaluated models, DeepSeek-ProverV2-7B and Goedel-Prover V2-8B achieve the highest overall scores, which is consistent with results reported in other benchmarks.

# 7 UNIT TEST ACCURACY

Evaluating agentic program synthesis requires more than compilable Lean code—it requires functions that behave correctly across diverse scenarios. We measure this with **unit test accuracy**, defined as the fraction of ground-truth tests passed (covering positive, negative, and edge cases). To compute this, we extract methods from generated Lean files, pair them with VeriBench unit tests by renaming method calls, and use the Lean compiler to verify outcomes (details in Appendix J). Table 2 shows that all agents outperform baseline prompting, with TRACE+ (self-debug) achieving the strongest overall accuracy. While Easy problems are handled well, CS and Real Code remain difficult; HumanEval falls in the mid-range, and Security sees modest but consistent gains from self-debug strategies. This provides an automated, end-to-end measure of functional reliability, complementing theorem-based evaluation.

| | Split | | | | | |
| Agent | Easy | CS | Real | HE | Security | Overall |
|---|---|---|---|---|---|---|
| Baseline Prompting | 0.317 | 0.000 | 0.400 | 0.616 | 0.572 | **0.486** |
| DSPY REACT | 0.317 | 0.350 | 0.400 | 0.393 | 0.576 | **0.432** |
| TRACE+ (Self-Debug) | 0.707 | 0.300 | 0.500 | 0.598 | 0.637 | **0.629** |
| TRACE++ (Self-Improve) | 0.573 | 0.200 | 0.400 | 0.554 | 0.659 | **0.568** |

Table 2: Average unit test accuracy on VERIBENCH. Columns correspond to benchmark splits: Easy Set (Easy), CS Set (CS), Real Python Code (Real), HumanEval Set (HE), and Security Set (Security). The rightmost column reports the overall mean across splits (bolded).

# 8 THEOREM SCORE QUALITY VERIBENCH WITH AN LLM JUDGE

**Motivation for LLM judge.** Evaluating end-to-end agentic autoformalization end-to-end is challenging because the agents must autonomously generate relevant theorems. In traditional autoformalization benchmark the models are given the exact natural language statement to formalize into a theorem, but in our setting, unless the property theorems are specified as property theorems, the agent must infer these property (theorems) on it's own. Therefore, we propose an LLM judge theorem to judge how similar the propose agent's formalization is to the gold reference. Our judge leverages Claude 3.7 and assigns an integer score between 0 (poor) and 10 (excellent) for each output file pair, evaluating correctness, completeness, and semantic alignment.

**Establishing Trust in LLM Judge.** To ensure trustworthiness of the LLM judge in evaluating agentic autoformalization, we systematically validate key properties the judge must satisfy in Figure 2: (1) scores decrease consistently as more theorems and tests are omitted, (2) scores similarly degrade when more implementation bugs are introduced, and (3) identical file pairs always yield maximal scores. Empirical results confirm these properties strongly hold (Pearson correlations up to -0.973), thus validating that our Claude 3.7-based LLM judge provides a reliable and semantically meaningful measure for correctness, completeness, and alignment of autonomously inferred theorems across VeriBench subsets (Table 3).

Table 3 presents the normalized average scores across different splits of VERIBENCH: Easy Set (Easy), CS Set (CS), Real Python Code (Real), HumanEval Set (HE), and Security Set (Sec). As expected, the baseline prompting approach achieves the lowest overall normalized score (0.470), reflecting its limited capability in autonomously identifying comprehensive theorem properties. Conversely, agentic methods leveraging iterative self-debugging and self-improvement demonstrate significant enhancements. Specifically, the DSPy ReAct agent, which incorporates feedback-driven

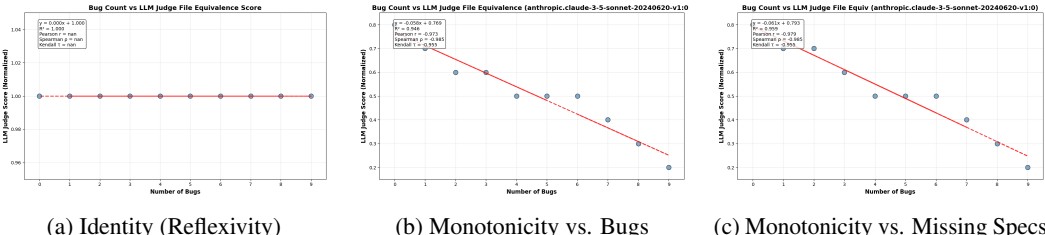

(a) Identity (Reflexivity)     (b) Monotonicity vs. Bugs     (c) Monotonicity vs. Missing Specs

Figure 2: **LLM Judge Trustworthiness Sanity Checks.** (a) Identical files receive the maximal score (consistency). (b) Scores decrease as more bugs are introduced (correctness sensitivity). (c) Scores decrease as specifications (theorems/tests) are removed (completeness sensitivity).

self-debugging, achieves the highest overall average score of 0.615. The Trace++ agent, employing both self-debugging and self-improvement mechanisms, also shows notable improvement, attaining an average score of 0.588. These results underscore the effectiveness of agentic, feedback-driven approaches in enhancing theorem synthesis quality compared to non-agentic baselines.

| | | Split | | | | | |
|---|---|---|---|---|---|---|---|
| **Agent** | **Eval** | **Easy** | **CS** | **Real** | **HE** | **Sec** | **Avg** |
| Baseline Prompting | Normalized | 0.580 | 0.690 | 0.472 | 0.410 | 0.347 | **0.470** |
| DSPy REACT | Normalized | 0.659 | 0.754 | 0.580 | 0.650 | 0.454 | **0.615** |
| TRACE+ (Self-Debug) | Normalized | 0.342 | 0.680 | 0.592 | 0.573 | 0.411 | **0.477** |
| TRACE++ (Self-Improve) | Normalized | 0.620 | 0.756 | 0.504 | 0.645 | 0.432 | **0.588** |

Table 3: **Theorem Quality Scores.** Average normalized scores in VERIBENCH split: Easy Set (Easy), CS Set (CS), Real Python Code (Real), HumanEval Set (HE), Security Set (Sec). All rows use Claude 3.7 as the agent model and as the LLM judge. Trace+ denotes the trace self-debug agent and Trace++ the self-debug and self-improve agent. The DSPy React agent only does self-debug. The LLM judge issues 5 scoring calls per file pair and returns an integer rubric score in between {0,1, ... , 10}. Normalized scores are computed as by dividing over the max score 10 and averaging across all the examples. The Avg Norm Score shows the overall average over the entire benchmark.

## 9 DISCUSSION AND CONCLUSION

**Limitations.** First, scope and construct validity: our gold specifications/theorems are rigorous yet necessarily incomplete; high scores indicate conformance to this specification and should not be over-interpreted as universal semantic correctness. Second, metric validity: compilation success, unit-test outcomes, and an LLM-based judgment jointly provide a fuller view than any single metric, but each remains a proxy — compilation can admit under-specification, tests sample behavior rather than exhaust it, and the judge may be sensitive to prompt framing. Third, extending coverage to additional libraries and harder "real-code" and security-critical splits will strengthen generality. Empirically, our results indicate that (i) proof synthesis remains the dominant bottleneck, (ii) feedback-driven agents outperform single-shot prompting under comparable budgets, and (iii) security/production-oriented tasks are substantially more challenging than toy problems. VERIBENCH verifies the translated Lean 4 artifact rather than the original Python source code; formally bridging this gap to provide end-to-end guarantees for the source program remains a key direction for future work.

**Future work** should broaden task diversity and library support, incorporate measures of proof readability and maintainability, and study sample-efficiency under strict budget regimes, including hybrid pipelines that combine symbolic search with agentic self-debugging. Taken together, these steps will promote progress toward verifiable AI coding systems.

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

## A  RELATED WORK (CONT.)

**Autoformalization.**  Complementing this line of work, Wang et al. (2024) present THEOREML-LAMA, a framework aimed at enhancing LLM translation into Lean 4. Drawing on over 100K proof examples from the Mathlib4 library, TheoremLlama employs a novel natural-language-to-formal-language (NL-FL) bootstrapping strategy and iterative proof synthesis. This enables the reuse of verified examples as templates for future translations. The framework achieves 36.48% and 33.61% accuracy on the MiniF2F-Valid and MiniF2F-Test benchmarks, respectively—surpassing GPT-4 by more than ten percentage points on both.

**Techniques for Code Verification.** IMPROVER (Ahuja et al., 2024) introduces a Lean-aware Chain-of-States prompting loop that integrates retrieval, best-of-$n$ sampling, and iterative correction to rewrite formal proofs with improved properties. By optimizing for metrics such as brevity and readability, ImProver reduces the number of tactics by half, doubles proof readability, and boosts theorem prover acceptance rates by over 80%. In addition, CLOVER (Sun et al., 2024) implements a closed-loop pipeline in which an LLM first generates code, docstrings, and formal annotations, then uses reconstruction-based prompting to enforce consistency across these outputs, and finally applies SMT-based verification to validate correctness. Evaluated on the CloverBench suite of Dafny programs, Clover accepts 87% of correct solutions, rejects 100% of flawed ones, and even uncovers bugs in human-written code—demonstrating the power of hybrid generation-verification pipelines. In a complementary direction, Zhou et al. (2025) improve general-purpose LLM-based graders by augmenting them with "privileged" information such as gold-standard solutions, grading rubrics, and detailed annotations. When necessary, the system provides targeted hints back to candidate models. This approach achieves grading performance on par with or exceeding that of specialized systems—and even expert humans—on difficult programming benchmarks.

**Agentic Frameworks and Tools for Code Verification.** TRACE (Cheng et al., 2024) proposes *generative optimization*, tuning entire computational workflows—including code, prompts, tool calls, and error signals—by treating execution traces as gradients in the OPTO framework. With a PyTorch-like API and the LLM-based optimizer OptoPrime, it supports diverse tasks such as prompt tuning, debugging, and robot control, rivaling specialized optimizers. Building on modular composition, DSPY (Khattab et al., 2024) treats LLM calls as declarative modules in a computational graph. Users define concise input–output signatures, and DSPy's compiler automatically bootstraps or fine-tunes pipelines using built-in "teleprompters." This enables a few-line programs to outperform expert-crafted prompts in math, QA, and agent workflows. Extending agentic capabilities to formal reasoning, PANTOGRAPH (Aniva et al., 2025) offers a programmatic interface to Lean 4 with support for advanced proof search. It exposes internal proof states and tactics for integration with learning agents, replacing human-facing interfaces with API-level control.

## B  DISCUSSION (CONT.)

**Why Lean 4?**  Lean 4 is a full fledge programming language and lets VeriBench contain full runnable code and machine-checked proofs in the same dependently-typed language. Ahead-of-time compilation with the Lake toolchain produces fast native binaries, while first-class `Task` primitives, async I/O, and a thread-pool scheduler enable genuine concurrent programs inside the prover. Lean's C-level foreign-function interface (FFI) lets those binaries call out to high-performance libraries when needed. On the proof side, Lean ships a powerful metaprogramming system written in Lean itself, giving researchers a programming interface to access its internals. This led to the creation of tools such as Pantograph and Aesop. Finally, the community-maintained `mathlib4` and `std4` libraries supply thousands of reusable theorems and data structures, and they are expanding quickly thanks to an active Zulip and GitHub ecosystem. Lean FRO also has plans to create libraries for verifying monadic programs. Lean 4 is backed by an unusually vibrant open-source community: hundreds of contributors refine `mathlib4` on GitHub each week, the public Zulip sees expert discussion around the clock, and even Fields-Medalist Terence Tao has chosen Lean to formalise portions of his current research—clear testimony to the ecosystem's accessibility and intellectual depth. Unlike the unit tests, fuzzers, and static-analysis pipelines common in industry—tools that sample inputs or rely on heuristics—Lean 4 supplies machine-checked proofs that a property holds for *all* executions. Its dependent type system can encode deep invariants (e.g., length-indexed arrays, bounded integers), so programs that violate them fail to compile, eliminating whole classes of bugs such as buffer overflows or integer wrap-around. Code, specification, and proof reside in the same file and are compiled by Lake into the shipped native binary, preventing the drift that arises when verification artifacts live outside the build. In short, Lean turns informal "best-effort" checks into formal, end-to-end guarantees without sacrificing performance or interoperability.

**Lean 4's limitations.**  Lean's toolchain is still younger than Coq's or Isabelle's, making its standard libraries and automation smaller, and thus some formalizations demand extra groundwork. While Lake delivers native executables, Lean's runtime has not been stress-tested at the scale of mainstream systems languages, meaning large-scale or safety-critical deployments may require additional vetting. Acknowledging these gaps clarifies that VeriBench chooses Lean 4 for its unique unified programming-plus-proving model and modern automation hooks, not because it already matches the decades-old industrial maturity of older theorem provers. *Static vs. runtime caveat:* Lean's guarantees are *static* and specification-relative: they certify the Lean program meets the stated properties, but they do not by themselves detect emergent runtime faults outside the model (e.g., I/O (Input/Output) failures, environment misconfiguration, FFI (Foreign Function Interface) unsoundness, resource exhaustion, or undefined behavior in linked C code). Such behaviors must either be modeled and proved, or mitigated with production safeguards (monitoring, sandboxing, input validation).

**Lean 4 versus Dafny.**  Unlike Dafny, whose verifier translates each program into the Boogie intermediate language and then discharges first-order verification conditions with an SMT solver such as Z3, Lean 4 reasons natively in a dependently-typed calculus. Because Lean 4's types can mention run-time data and the very same source code is ahead-of-time compiled to a native binary via Lake, we can both state and prove value-indexed, higher-order properties (e.g., length-indexed vectorsand run the verified program itself—an end-to-end, fidelity that Dafny's SMT-centred, Boogie-to-Z3 workflow cannot natively match. This gives Lean the expressive power to specify and prove higher-order, data-dependent properties—precisely the kind of semantic guarantees VeriBench seeks to test—while still yielding runnable binaries compiled by the same toolchain. Dafny's SMT-centric workflow offers impressive push-button automation for imperative code but cannot natively encode the richer specifications (e.g., length-indexed vectors, algebraic invariants) that Lean handles directly.

**Lean 4 versus TLA+.**  TLA+ excels at high-level specification of concurrent and distributed protocols, with correctness checked by the TLC model-checker and the TLAPS proof system that dispatches first-order obligations to external provers. However, TLA+ specifications are not executable programs; a separate implementation step is required, and state-space explosion can limit model-checking scalability. VeriBench instead needs a prover where the specification, proof, and runnable code live in the same language. Lean 4's dependently-typed core lets us capture fine-grained, data-dependent invariants and then compile the very same artifacts to fast native binaries—capabilities outside TLA+'s scope.

| Model | VeriBench | Alpaca |
|---|---|---|
| LLaMA 3.1–8B (DSPy) | 0/8 | 1/40 |
| LLaMA 70B (DSPy) | 0/8 | 2/40 |
| WattAI 70B (DSPy) | 0/8 | 5/40 |
| Claude 3.5 Sonnet v1 (DSPy) | 8/8 | 24/40 |
| Claude 3.5 Sonnet v2 (DSPy) | 4/8 | 1/40 |
| Claude 3.7 Sonnet (DSPy) | 1/8 | 7/40 |
| Claude 3.5 Sonnet v1 (Baseline) | 8/116 | 22/40 |
| Claude 3.5 + Self-Debug + Judge (Trace) | 42/116 | — |

Table 4: Number of problems resolved by language models on Mini-VeriBench and Alpaca (Gupta, 2023) benchmarks.

| Model | HumanEval | EasySet | CSSet | SecuritySet | Average |
|---|---|---|---|---|---|
| Baseline | 4/51 | 3/41 | 0/10 | 1/11 | 8/113 |
| Trace v1 | 31/51 | 30/41 | 3/10 | 4/11 | 67/113 |
| Trace v2 | 8/51 | 24/41 | 1/10 | 7/11 | 40/113 |

Table 5: Performance on VeriBench (in %). For Trace, we used Claude 3.5 v1. For the baseline, we called Claude 3.5 v1 without access to the compiler and LLM judge feedback. "Trace v1" means just self-debug by the language models, while "Trace v2" indicates self-debug plus LLM-judge.

## C    VERIBENCH-V0 RESULTS

### C.1    MINI-VERIBENCH RESULTS

Table 4 We evaluate a range of prominent language models on VeriBench, including both open-source and proprietary systems. Specifically, our evaluation covers LLaMA 3.1–8B, LLaMA 70B, WattAI (LLaMA 70B trained for tool use), Claude 3.5 Sonnet, Claude 3.5 Sonnet (v1 and v2), and Claude 3.7. This diverse selection provides a balanced view of current model capabilities in code translation to formal code under different inference configurations.

#### C.1.1    MAIN RESULTS VERIBENCH-V0.1

Table 4 summarizes the compilation success across multiple settings and models, including Trace-agents variants. These results highlight the challenges of formal code verification via LLMs and the benefits of reward-guided, self-optimizing agent architectures.

Despite comprising just 8 HumanEval translations, current frontier models demonstrate significant limitations: Claude 3.7 Sonnet achieves compilation on only 1 of 8 test examples (12.5%), while Claude 3.5 Sonnet achieves compilation on only 4 of 8 tasks (50%), and LLaMA-70B fails to compile any programs in the Lean 4 HumanEval subset, even with 50 feedback-guided attempts. Note on the easier Alpaca Code subset, the WattAI 70B (a LaLMA 70B tool use model) got barely 11 out of 40 compilation successes (27.5%).

Notably, of the approaches evaluated, our experiments reveal that only a self-optimizing agent architecture achieves meaningful compilation rates approaching 90%. VeriBench establishes a rigorous foundation for developing AI systems capable of synthesizing provably correct, bug-free code, thereby advancing the trajectory toward more secure and dependable software infrastructure.

## D    FUTURE DIRECTIONS

We envision VeriBench as a launching pad for multilingual code verification. Many HumanEval problems exist in MultiPL-E's corpus of 47 languages, allowing future extensions from Python to C, C++, Rust, OCaml, and more. This opens the door for evaluating how models generalize verification strategies across language boundaries.

# E  WHY VERIBENCH DOES *not* PROMISE A "COMPLETE" THEOREM LIST

## E.1  MOTIVATION

It is tempting to publish VeriBench together with a fixed theorem set $\Sigma$ and claim:

> *"If your tool can derive every statement in $\Sigma$, then the analysed program is completely correct."*

In practice that promise is unattainable once programs are written in a Turing-complete language like Python, C, and Java. The obstacle is not a lack of ingenuity but an established negative result from computability theory.

## E.2  RICE'S THEOREM

[Rice, 1953, informal] Let $P$ range over all programs in a Turing-complete language, and let $\mathcal{S}$ be a property that is

1. **semantic**: depends only on the input–output behaviour of $P$,
2. **non-trivial**: is true for some programs and false for others.

Then no algorithm can decide for *every* program whether it satisfies $\mathcal{S}$.

Thus, already for the Halting Problem—"does $P$ terminate on input $x$?"—a universal decision procedure cannot exist; Rice's theorem extends this impossibility to *any* meaningful behaviour-oriented property.

## E.3  IMPACT ON A "COMPLETE" THEOREM LIST

Assume, for contradiction, that a recursively enumerable and sound collection $\Sigma$ captured *every* true semantic statement about a reference program $P$. Because $\Sigma$ is enumerable, we could mechanically search its proof stream: the first appearance of either $\psi$ or $\neg\psi$ would decide $\psi$. For many interesting $\psi$ (e.g. total termination, sorted output) this would yield a universal algorithm contradicting Rice's theorem. Therefore any practical, machine-enumerable $\Sigma$ must remain incomplete—some true statements are unavoidably absent.

## E.4  VERIBENCH DESIGN CHOICE

Rice's theorem bars us from publishing a *single* theory that is both recursively enumerable and complete for *all* benchmarks. Yet many benchmarks in practice—especially mathematically-defined, total algorithms over finite or structurally decreasing data—*do* admit a fully exhaustive specification. VeriBench therefore pursues **instance-level completeness** by structuring the gold lean 4 reference files in a rigorous manner as stated in Appendix F and following two-step human curation processes.

# F  GOLD STANDARD FOR LEAN 4 BENCHMARK FILES

**Purpose.**  Each gold file specifies the intended Lean implementation by: (i) showing functional and (optionally) imperative code; (ii) declaring a *Pre-condition* predicate; (iii) stating algebraic/semantic properties as separate theorems; (iv) defining the *Post-condition* as their conjunction; (v) proving a *Correctness* theorem ($\rightarrow$) by combining those theorems; and (vi) if an imperative variant exists, proving an *Equivalence* theorem between functional and imperative code. This design supports partial-credit grading and consistent comparison even under different theorem factorizations.

**Required order.**

1. **Implementation.** Functional definition of $Prog$.
2. **Unit tests.** Include edge, positive, and negative cases: – Positive: valid input–output pairs satisfying *Pre* and all properties. – Negative: inputs/claims violating *Pre* or *Post* (e.g., fabricated equalities, out-of-bounds, overflow).

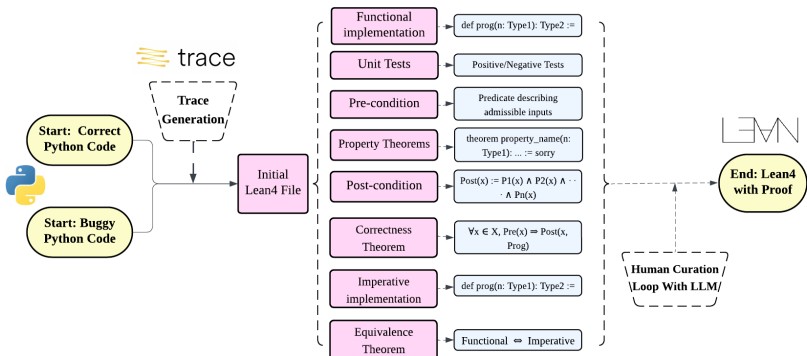

Figure 3: Benchmark Creation Procedure and Structure.

3. **Pre-condition.** Predicate : *Input* →. Use ≡ True if the type already enforces constraints. No proof required.

4. **Properties.** Each property (e.g., identity, commutativity, associativity, distributivity, safety) stated as its own theorem—these are the atomic proof obligations.

5. **Post-condition.** Conjunction of properties, preserving order:

$$Post(x) := P_1(x) \land P_2(x) \land \cdots \land P_n(x).$$

6. **Correctness.** For all valid $x$, $(x) \Rightarrow Post(x, Prog)$. Proof is a direct combination of the property theorems.

7. **Imperative equivalence.** Provide an imperative version, add tests, and prove equivalence with the functional implementation.

## G   CONSTRUCTION PIPELINE AND VALIDITY GUARANTEES

A common objection to LLM–generated benchmarks states that the model which creates the data may already have the capability to solve, compromising test integrity and more (**?**). We address this concern directly.

**Manual audit with public provenance.**   After o3 drafts each Lean4 artefact, a curator opens a GitHub issue that is inspected by a 2nd human reviewers, edits the patch when needed, and merges only after the issues resolved (e.g., no comprehensive set of theorems). Every change, comment, and decision is preserved in the repository history, providing reproducible evidence of human oversight.

**Kernel-enforced correctness.**   A pull request must compile to be accepted. Because the Lean kernel is a proof checker, compilation implies that every implementation, unit test, theorem, and proof is logically sound. Frontier models struggle to compile at 59therefore the final tasks necessarily exceed the generator's capabilities.

**No leakage of final solutions.**   Curator edits routinely alter types, theorem statements, or proof strategies—changes the originating LLM cannot anticipate. The published tasks thus differ from the raw LLM output and are not trivially solvable by the same model.

**Benchmark Leaderboard Rankings are Robust to noise in Benchmarks.**   Model *rankings* are stable even on noisy datasets (**?**). Therefore it is known that imperfections would not distort model rankings.

Key benefits of VeriBench Lean Gold File structure.

**Key benefits of the structure.**

- **Granular proof evaluation.** Splitting the post-condition into separate theorems provides independent proof obligations and partial credit when only a subset is generated or proved.

- **Robust equivalence via a summary.** Even if a generated file formats or partitions properties differently from the gold file, its *Correctness* theorem (the conjunction of its own property theorems) offers a uniform target. The judge checks that the candidate's bundled statement implies (or is equivalent to) the gold ($Post$), recognizing semantically correct but syntactically different decompositions.

- **Clear separation of roles. Pre** is a declarative predicate ($Prop$) stating admissible inputs. Each post-condition component is a **theorem** requiring proofs. *Post* is the ordered conjunction of these properties; **Correctness** bundles their proofs.

- **Imperative cross-check.** Providing an imperative implementation and an *Equivalence* theorem strengthens trust: it guards loop→recursion translation errors, enables testing on a realistic stateful version, and lets evaluators accept candidates whose functional code differs internally as long as it is provably equivalent to the imperative reference.

## H    STANDARD FOR CORRECT PYTHON INPUT FILES

**Scope.**    Each correct Python file accompanies the Lean gold file with a simple, executable reference. It declares a pre-condition, enforces it *at the start* of the implementation, and includes positive/negative unit tests.

**Required structure (in order).**

1. **Docstring.** One–two sentences describing the function's intent and edge cases.

2. **Python pre-condition.** A pure predicate `pre(...)` → `bool` encoding admissible inputs.

3. **Implementation with early pre-check.** `prog(...)` must call `pre(...)` first and *raise AssertionError* if it fails; otherwise perform the computation.

4. **Unit tests.** A small suite with:

    - *Positive tests*: valid inputs with expected outputs (include edge cases).

    - *Negative tests*: invalid inputs that must raise `AssertionError`.

# I  UNIT TEST ACCURACY

| Agent | Eval | Split | | | | | Overall |
|---|---|---|---|---|---|---|---|
| | | **Easy** | **CS** | **Real** | **HE** | **Security** | |
| DSPY REACT | Functional | 0.366 | 0.400 | 0.000 | 0.393 | 0.364 | 0.407 |
| | Imperative | 0.268 | 0.300 | 0.800 | 0.393 | 0.788 | 0.457 |
| | **Average** | **0.317** | **0.350** | **0.400** | **0.393** | **0.576** | **0.432** |
| Trace Baseline | Functional | 0.341 | 0.000 | 0.000 | 0.607 | 0.273 | 0.429 |
| | Imperative | 0.293 | 0.000 | 0.800 | 0.625 | 0.871 | 0.543 |
| | **Average** | **0.317** | **0.000** | **0.400** | **0.616** | **0.572** | **0.486** |
| TRACE+ (Self-Debug) | Functional | 0.756 | 0.300 | 0.200 | 0.589 | 0.318 | 0.586 |
| | Imperative | 0.659 | 0.300 | 0.800 | 0.607 | 0.955 | 0.671 |
| | **Average** | **0.707** | **0.300** | **0.500** | **0.598** | **0.637** | **0.629** |
| TRACE++ (Self-Improve) | Functional | 0.610 | 0.200 | 0.000 | 0.536 | 0.364 | 0.521 |
| | Imperative | 0.537 | 0.200 | 0.800 | 0.571 | 0.955 | 0.614 |
| | **Average** | **0.573** | **0.200** | **0.400** | **0.554** | **0.659** | **0.568** |

Table 6: Consolidated unit test accuracy on VERIBENCH. Rows group agents, and within each agent, we report *Functional*, *Imperative*, and their *Average*. Columns correspond to benchmark splits: Easy Set (Easy), CS Set (CS), Real Python Code (Real), HumanEval Set (HE), and Security Set (Security). The rightmost column reports the overall mean across splits for each evaluation type.

# J  APPENDIX: JUDGING METHODS

Our evaluation pipeline has three simple stages: (1) *extract* the intended solution function(s) from each generated Lean file; (2) *align* VeriBench's unit tests to those functions by renaming calls as needed; and (3) *verify* with the Lean compiler whether each test passes. We implement two extraction strategies: a lightweight heuristic parser and a more robust LLM-based extractor. We stress-test both approaches on original files, on versions with obfuscated identifiers, and on versions with small induced errors. Across these settings, the LLM-based judge is more reliable and roughly $5\times$ faster to stand up in practice. The result is a scalable, black-box evaluation of agentic program synthesis that focuses on behavior—not just compilability—via unit-test accuracy.

# K  MORE SIMPLIFIED BENCHMARK EXAMPLES

In this section we provide more simplified examples from VERIBENCH.

```python
# Implementation
def my_max(a: int, b: int) -> int:
    """
    Return the larger of two non-negative integers.
    """
    return b if a <= b else a

# Tests
from typing import Callable

def check(candidate: Callable[[int, int], int]) -> bool:
    assert candidate(7, 3) == 7, f"expected 7 from (7,3) but got {candidate(7, 3)}"
    ... (other tests) ...
    return True

if __name__ == "__main__":
    assert check(my_max), f"Failed: {__file__}"
    print(f'All tests passed: {__file__}!')
```

Listing 3: An exemplar input Python code of VeriBench-EasySet (simplified for showcase).

```
namespace MyMax
```

```
-- Implementation
def myMax (a b : Nat) : Nat :=
 if _ : a ≤ b then b else a

infixl:70 "⊔"=> myMax -- left-associative, precedence 70

-- Tests
#eval myMax 7 3 -- expect 7
example : myMax 7 3 = 7 := by native_decide
#eval myMax 0 0 -- expect 0
example : myMax 0 0 = 0 := by native_decide

-- Theorems
@[simp] theorem max_left_identity (n : Nat) : myMax 0 n = n := sorry

-- Theorem Right Identity
@[simp] theorem max_right_identity (n : Nat) : myMax n 0 = n := sorry

... (other theorems) ...

end MyMax
```

Listing 4: An exemplar golden output Lean 4 code of VeriBench-EasySet.

```
# Implementation
from typing import List, Optional
def binary_search(arr: List[int], target: int) -> Optional[int]:
    """
    Binary search implementation that searches for a target value in a sorted list.
    Returns the index if found, None if not found.
    """
    if not arr:
        return None

    left, right = 0, len(arr) - 1

    while left <= right:
        mid = (left + right) // 2
        mid_val = arr[mid]

        if mid_val == target:
            return mid
        elif mid_val < target:
            left = mid + 1
        else:
            right = mid - 1

    return None

# Tests
from typing import Callable
def check(candidate: Callable[[List[int], int], Optional[int]]) -> bool:
    assert candidate([1, 2, 3, 4, 5], 1) == 0
    assert candidate([1, 2, 3, 4, 5], 3) == 2
    ... (more tests) ...

    print("Pass: all correct!")
    return True

if __name__ == "__main__":
    assert check(binary_search), f"Failed: {__file__}"
```

Listing 5: An exemplar input Python code of VeriBench-CSSet (simplified for showcase).

```
import Mathlib.Data.List.Sort
import Mathlib.Data.List.Basic

namespace BinarySearch
open List

-- Implementation
partial def binarySearchAux (arr : List Nat) (target : Nat) (left right : Nat) : Option Nat :=
 if left > right then
   none
 else
   let mid := (left + right) / 2
   if mid >= arr.length then
```

```
     none
   else
     let midVal := arr.get <mid, by sorry>
     if midVal = target then
       some mid
     else if midVal < target then
       binarySearchAux arr target (mid + 1) right
     else
       binarySearchAux arr target left (mid - 1)

def binarySearch (arr : List Nat) (target : Nat) : Option Nat :=
  if arr.isEmpty then
    none
  else
    binarySearchAux arr target 0 (arr.length - 1)

/-- Linear search for comparison and verification -/
def linearSearch (arr : List Nat) (target : Nat) : Option Nat :=
  arr.findIdx? (· = target)

-- Theorem: If binarySearch returns Some i, then arr[i] = target
theorem correctness_binarySearch (arr : List Nat) (target : Nat) (i : Nat) :
  binarySearch arr target = some i →arr[i]? = some target := by
  sorry

-- Theorem: If target is in the sorted array, then binarySearch finds it
theorem completeness_binarySearch (arr : List Nat) (target : Nat) :
  List.Sorted (fun x y => x ≤y) arr →target ∈arr →
  ∃i, binarySearch arr target = some i := by
  sorry

... (more theorems) ...

end BinarySearch
```

Listing 6: An exemplar golden output Lean 4 code of VeriBench-CSSet (simplified for showcase).

```python
# Implementation
def unsafe_copy(dst: bytearray, src: bytearray) -> None:
    """
    Copy bytes from `src` into `dst` at the same indices, without any bounds checking.
    If `len(src) > len(dst)`, this will raise an IndexError (buffer overflow).
    """
    for i, b in enumerate(src):
        dst[i] = b

# Tests
def check(candidate) -> bool:
    # 1) Safe copy: src fits in dst
    d = bytearray(3)
    s = bytearray(b'abc')
    candidate(d, s)
    assert bytes(d) == b'abc'

    ... (other tests) ...

    return True

assert check(unsafe_copy), "Candidate failed buffer-overflow tests"
print("Pass!")
```

Listing 7: An exemplar input Python code of VeriBench-SecuritySet (simplified for showcase).

```
namespace BufferOverflow

-- Implementation
def unsafeCopy (dst src : List UInt8) : Option (List UInt8) :=
  let n := dst.length
  -- fold over enumerated bytes with their indices
  src.enum.foldl (fun o (i, b) =>
    o.bind fun acc =>
    if h : i < n then
      some (acc.set i b)
    else
      none
  ) (some dst)

-- Tests
```

```
example : unsafeCopy [0, 0, 0] [1,2] = some [1,2,0] := by rfl
example : unsafeCopy [0, 0] [1,2,3] = none := by rfl

... (other tests) ...

-- Theorem: safety precondition
theorem copy_safe {dst src : List UInt8}
  (h : src.length ≤ dst.length) :
  ∃newDst, unsafeCopy dst src = some newDst := by
  unfold unsafeCopy

-- Theorem: overflow detection
theorem copy_overflow {dst src : List UInt8}
  (h : dst.length < src.length) :
  unsafeCopy dst src = none := by
  unfold unsafeCopy
  admit

end BufferOverflow
```

Listing 8: An exemplar golden output Lean 4 code of VeriBench-SecuritySet (simplified for showcase).

```
# -- Implementation --
# source: https://github.com/python/cpython/blob/3.13/Lib/heapq.py

# 'heap' is a heap at all indices >= startpos, except possibly for pos.
# Restore the heap invariant by bubbling the new item up.
def _siftdown(heap, startpos, pos):
    newitem = heap[pos]
    while pos > startpos:
        parentpos = (pos - 1) >> 1
        parent = heap[parentpos]
        if newitem < parent:
            heap[pos] = parent
            pos = parentpos
            continue
        break
    heap[pos] = newitem

def heappush(heap, item):
    """Push item onto heap, maintaining the heap invariant."""
    heap.append(item)
    _siftdown(heap, 0, len(heap) - 1)

# -- Tests (simplified) --
from typing import Callable

def _check_invariant(h):
    for i in range(1, len(h)):
        p = (i - 1) >> 1
        assert h[p] <= h[i], "heap invariant violated"

def check(candidate: Callable[[list, int], None]) -> bool:
    # Basic sanity
    h = []
    candidate(h, 3); _check_invariant(h)
    candidate(h, 1); _check_invariant(h)
    candidate(h, 2); _check_invariant(h)
    assert h[0] == 1

    # Equal elements
    h = []
    candidate(h, 5); candidate(h, 5); _check_invariant(h)
    assert h[0] == 5

    # ... (additional randomized/property tests omitted) ...
    return True

assert check(heappush)
```

Listing 9: An exemplar input Python code of VeriBench-RealCode (simplified for showcase).

```
/-!
# VeriBench  Heappush (min-heap, simplified)
```

```
Order:
1. Implementation ('_siftdown', 'heappush')
2. Minimal checks
3. Pre-condition
4. Properties
5. Post-condition
6. Correctness
-/

namespace HeapPush

/-- Move the leaf at 'pos' up until the min-heap invariant holds. -/
def _siftdown (heap : Array Int) (startpos pos : Nat) : Array Int :=
  let newitem := heap[pos]!
  let rec loop (h : Array Int) (pos : Nat) : Array Int :=
    if pos > startpos then
      let parentpos := (pos - 1) >>> 1
      let parent := h[parentpos]!
      if newitem < parent then
        loop (h.set! pos parent) parentpos
      else
        h.set! pos newitem
    else
      h.set! pos newitem
  loop heap pos

/-- Push an element and restore the heap invariant. -/
def heappush (heap : Array Int) (item : Int) : Array Int :=
  let h := heap.push item
  _siftdown h 0 (h.size - 1)

/-! ## Minimal checks (others omitted) -/
example : (heappush (heappush (heappush (#[] : Array Int) 3) 1) 2)[0]! = 1 := by native_decide
example :
  let h1 := heappush (#[] : Array Int) 5
  let h2 := heappush h1 5
  h1[0]! = 5 ∧ h2[0]! = 5 := by native_decide

/-- Boolean heap invariant: for every child, parent ≤ child. -/
def checkInvariant (h : Array Int) : Bool :=
  let n := h.size
  let rec go (i : Nat) : Bool :=
    if i ≥ n then true else
    if i = 0 then go (i+1) else
    let p := (i - 1) >>> 1
    (h[p]! ≤ h[i]!) && go (i+1)
  go 0

/-! ## Pre-condition -/
def Pre (heap : Array Int) : Prop :=
  checkInvariant heap = true

/-! ## Properties -/
def prop_invariant (heap : Array Int) (item : Int) : Prop :=
  checkInvariant (heappush heap item) = true

def prop_size (heap : Array Int) (item : Int) : Prop :=
  (heappush heap item).size = heap.size + 1

def prop_multiset (heap : Array Int) (item : Int) : Prop :=
  List.Perm (heappush heap item).toList (item :: heap.toList)

/-! ## Theorems (sketched; proofs omitted) -/
@[simp] theorem invariant_thm (heap : Array Int) (item : Int) (hPre : Pre heap) :
  prop_invariant heap item := by
  sorry

@[simp] theorem size_thm (heap : Array Int) (item : Int) :
  prop_size heap item := by
  sorry

@[simp] theorem multiset_thm (heap : Array Int) (item : Int) :
  prop_multiset heap item := by
  sorry

/-! ## Post-condition -/
def Post_prop (heap : Array Int) (item : Int) : Prop :=
  prop_invariant heap item ∧ prop_size heap item ∧ prop_multiset heap item

/-! ## Correctness -/
theorem correctness_thm (heap : Array Int) (item : Int) (hPre : Pre heap) :
```

```
  Post_prop heap item := by
  exact And.intro
    (invariant_thm heap item hPre)
    (And.intro (size_thm heap item) (multiset_thm heap item))

end HeapPush
```

Listing 10: An exemplar golden output Lean 4 code of VeriBench-RealCode (simplified for showcase).

## L    FULL BENCHMARK EXAMPLES

In this section we provide full examples from VERIBENCH.

```python
# -- Implementation --

def my_max(a: int, b: int) -> int:
    """
    Return the larger of two non-negative integers.

    >>> my_max(7, 3)
    7
    >>> my_max(0, 0)
    0
    """
    return b if a <= b else a

# -- Tests --
from typing import Callable

def check(candidate: Callable[[int, int], int]) -> bool:
    print(f'Running tests for {candidate.__name__}...')
    # Basic unit tests
    assert candidate(7, 3) == 7, f"expected 7 from (7,3) but got {candidate(7, 3)}"
    # Edge unit tests
    assert candidate(0, 0) == 0, f"expected 0 from (0,0) but got {candidate(0, 0)}"

    # Property checks on a small domain
    for x in range(6):
        # idempotence
        assert candidate(x, x) == x, f"idempotence violated for {x}"

        for y in range(6):
            # commutativity
            lhs = candidate(x, y)
            rhs = candidate(y, x)
            assert lhs == rhs, (
                f"commutativity violated for ({x},{y}): {lhs} != {rhs}"
            )

            # upper-bound property
            assert x <= candidate(x, y), f"left bound violated for ({x},{y})"
            assert y <= candidate(x, y), f"right bound violated for ({x},{y})"

            for z in range(6):
                # associativity
                left_assoc = candidate(candidate(x, y), z)
                right_assoc = candidate(x, candidate(y, z))
                assert left_assoc == right_assoc, (
                    "associativity violated for "
                    f"({x},{y},{z}): {left_assoc} != {right_assoc}"
                )

    return True

if __name__ == "__main__":
    assert check(my_max), f"Failed: {__file__}"
    print(f'All tests passed: {__file__}!')
```

Listing 11: An exemplar input Python code of VeriBench-EasySet.

```
/-!
# Implementation
-/

namespace MyMax
/--
```

```
Implementation of my custom maximum function.

`myMax a b` returns the larger of two natural numbers.

## Examples

#eval myMax 7 3 -- expect 7
#eval myMax 0 0 -- expect 0
-/
def myMax (a b : Nat) : Nat :=
  if _ : a ≤ b then b else a

infixl:70 "⊔" => myMax -- left-associative, precedence 70

/-!
# Tests
We use `#eval` to print results, then nameless `example` to confirm correctness
(especially in cases where `native_decide` is used to prove the example).
-/

-- Functional tests
#eval myMax 7 3 -- expect 7
example : myMax 7 3 = 7 := by native_decide

#eval myMax 0 0 -- expect 0
example : myMax 0 0 = 0 := by native_decide

/-!
# Theorems
-/

/-- Theorem Left Identity: Taking max with zero on the left acts as the identity. -/
@[simp] theorem max_left_identity (n : Nat) : myMax 0 n = n := sorry

/-- Theorem Right Identity: Taking max with zero on the right acts as the identity. -/
@[simp] theorem max_right_identity (n : Nat) : myMax n 0 = n := sorry

/-- Theorem Commutativity: The order of the arguments does not affect the maximum. -/
@[simp] theorem max_commutativity (a b : Nat) : myMax a b = myMax b a := sorry

/-- Theorem Idempotence: Taking max of a number with itself yields that number. -/
@[simp] theorem max_idempotent (a : Nat) : myMax a a = a := sorry

/-- Theorem Left Bound: The first argument never exceeds the maximum. -/
theorem max_left_bound (a b : Nat) : a ≤ myMax a b := sorry

/-- Theorem Right Bound: The second argument never exceeds the maximum. -/
theorem max_right_bound (a b : Nat) : b ≤ myMax a b := sorry

/--
Imperative implementation of `myMax`.

`myMaxImp a b` computes the same maximum using mutable state:
start with `m := a`, then overwrite with `b` if `b` is larger.

## Examples

#eval myMaxImp 7 3 -- expect 7
#eval myMaxImp 0 0 -- expect 0
-/
def myMaxImp (a b : Nat) : Nat :=
  Id.run do
    let mut m : Nat := a
    for x in [a, b] do -- loop over both inputs
      if m ≤ x then
        m := x
    return m

-- Imperative tests
#eval myMaxImp 7 3 -- expect 7
example : myMaxImp 7 3 = 7 := by native_decide

#eval myMaxImp 0 0 -- expect 0
example : myMaxImp 0 0 = 0 := by native_decide

/--
Theorem Equivalence of Functional and Imperative Maximum:
Both implementations produce identical results for all inputs.
-/
theorem myMax_equiv_functional_imperative (a b : Nat) :
  myMax a b = myMaxImp a b := sorry
```

```
1296
1297    end MyMax
1298
```

Listing 12: An exemplar golden output Lean 4 code of VeriBench-EasySet.

```
1300
1301    # -- Implementation --
        from typing import List, Optional
1302
        def binary_search(arr: List[int], target: int) -> Optional[int]:
1303        """
1304        Binary search implementation that searches for a target value in a sorted list.
           Returns the index if found, None if not found.
1305
1306        >>> binary_search([1, 2, 3, 4, 5], 3)
           2
1307        >>> binary_search([1, 2, 3, 4, 5], 6)
1308
1309        >>> binary_search([], 1)
1310
           """
1311        if not arr:
              return None
1312
1313        left, right = 0, len(arr) - 1
1314        while left <= right:
              mid = (left + right) // 2
1315           mid_val = arr[mid]
1316
              if mid_val == target:
1317              return mid
1318           elif mid_val < target:
                 left = mid + 1
1319           else:
1320              right = mid - 1
1321        return None
1322
        # -- Tests --
1323    from typing import Callable
        def check(candidate: Callable[[List[int], int], Optional[int]]) -> bool:
1324        # Basic functionality tests
           assert candidate([1, 2, 3, 4, 5], 1) == 0
1325        assert candidate([1, 2, 3, 4, 5], 3) == 2
1326        assert candidate([1, 2, 3, 4, 5], 5) == 4
           assert candidate([1, 2, 3, 4, 5], 6) is None
1327        assert candidate([1, 2, 3, 4, 5], 0) is None
1328
           # Edge cases
1329        assert candidate([], 1) is None
1330        assert candidate([5], 5) == 0
           assert candidate([5], 3) is None
1331
1332        # Larger arrays
           assert candidate([1, 3, 5, 7, 9], 3) == 1
1333        assert candidate([1, 3, 5, 7, 9], 7) == 3
1334        assert candidate([1, 3, 5, 7, 9], 4) is None
           assert candidate([10, 20, 30, 40, 50, 60], 60) == 5
1335        assert candidate([10, 20, 30, 40, 50, 60], 10) == 0
1336
           # Test with duplicates (binary search may return any valid index)
1337        test_arr = [1, 2, 3, 3, 3, 4, 5]
1338        result = candidate(test_arr, 3)
           assert result is not None and test_arr[result] == 3 and 2 <= result <= 4
1339
1340        # Large sorted array test
           large_arr = list(range(100))
1341        assert candidate(large_arr, 49) == 49
1342        assert candidate(large_arr, 99) == 99
           assert candidate(large_arr, 100) is None
1343
1344        # Two element arrays
           assert candidate([1, 2], 1) == 0
1345        assert candidate([1, 2], 2) == 1
1346        assert candidate([1, 2], 3) is None
1347
1348        print("Pass: all correct!")
           return True
1349
        if __name__ == "__main__":
```

```
    assert check(binary_search), f"Failed: {__file__}"
```

Listing 13: An exemplar input Python code of VeriBench-CSSet.

```
import Mathlib.Data.List.Sort
import Mathlib.Data.List.Basic

/-!
# Implementation
-/

namespace BinarySearch

open List

/-- Binary search implementation using recursive approach with bounds -/
partial def binarySearchAux (arr : List Nat) (target : Nat) (left right : Nat) : Option Nat :=
  if left > right then
    none
  else
    let mid := (left + right) / 2
    if mid >= arr.length then
      none
    else
      let midVal := arr.get <mid, by sorry>
      if midVal = target then
        some mid
      else if midVal < target then
        binarySearchAux arr target (mid + 1) right
      else
        binarySearchAux arr target left (mid - 1)

/-- Binary search that searches for a target value in a sorted list.
    Returns Some index if found, None if not found. -/
def binarySearch (arr : List Nat) (target : Nat) : Option Nat :=
  if arr.isEmpty then
    none
  else
    binarySearchAux arr target 0 (arr.length - 1)

/-- Linear search for comparison and verification -/
def linearSearch (arr : List Nat) (target : Nat) : Option Nat :=
  arr.findIdx? (· = target)

/-!
# Theorems
-/

/--
**Correctness**: If `binarySearch` returns `Some i`, then `arr[i] = target`.
-/
theorem correctness_binarySearch (arr : List Nat) (target : Nat) (i : Nat) :
  binarySearch arr target = some i →arr[i]? = some target := by
  sorry

/--
**Completeness**: If `target` is in the sorted array, then `binarySearch` finds it.
-/
theorem completeness_binarySearch (arr : List Nat) (target : Nat) :
  List.Sorted (fun x y => x ≤y) arr →target ∈arr →
  ∃i, binarySearch arr target = some i := by
  sorry

/--
**Valid Index**: If `binarySearch` returns `Some i`, then `i` is a valid index.
-/
theorem valid_index_binarySearch (arr : List Nat) (target : Nat) (i : Nat) :
  binarySearch arr target = some i →i < arr.length := by
  sorry

/--
**Not Found**: If `binarySearch` returns `None`, then `target` is not in the array
(assuming the array is sorted).
-/
theorem not_found_binarySearch (arr : List Nat) (target : Nat) :
  List.Sorted (fun x y => x ≤y) arr →
  binarySearch arr target = none →target ∉arr := by
  sorry
```

```
1404   /--
1405   **Equivalence with Linear Search**: On sorted arrays, binary search and linear search
1406   find the same elements (though possibly different indices for duplicates).
       -/
1407   theorem equiv_linearSearch_binarySearch (arr : List Nat) (target : Nat) :
1408     List.Sorted (fun x y => x ≤ y) arr →
         (binarySearch arr target).isSome ↔ (linearSearch arr target).isSome := by
1409     sorry
1410
       /-!
1411   # Imperative Tests
1412   -/
1413   /-- expected: some 0 -/
1414   example : binarySearch [1, 2, 3, 4, 5] 1 = some 0 := by native_decide
       #eval binarySearch [1, 2, 3, 4, 5] 1 -- expected: some 0
1415
1416   /-- expected: some 2 -/
       example : binarySearch [1, 2, 3, 4, 5] 3 = some 2 := by native_decide
1417   #eval binarySearch [1, 2, 3, 4, 5] 3 -- expected: some 2
1418
1419   /-- expected: some 4 -/
       example : binarySearch [1, 2, 3, 4, 5] 5 = some 4 := by native_decide
1420   #eval binarySearch [1, 2, 3, 4, 5] 5 -- expected: some 4
1421
1422   /-- expected: none -/
       example : binarySearch [1, 2, 3, 4, 5] 6 = none := by native_decide
1423   #eval binarySearch [1, 2, 3, 4, 5] 6 -- expected: none
1424   /-- expected: none -/
       example : binarySearch [1, 2, 3, 4, 5] 0 = none := by native_decide
1425   #eval binarySearch [1, 2, 3, 4, 5] 0 -- expected: none
1426
1427   /-- expected: none -/
       example : binarySearch [] 1 = none := by native_decide
1428   #eval binarySearch [] 1 -- expected: none
1429
1430   /-- expected: some 0 -/
       example : binarySearch [5] 5 = some 0 := by native_decide
1431   #eval binarySearch [5] 5 -- expected: some 0
1432   /-- expected: none -/
       example : binarySearch [5] 3 = none := by native_decide
1433   #eval binarySearch [5] 3 -- expected: none
1434
1435   /-- expected: some 1 -/
       example : binarySearch [1, 3, 5, 7, 9] 3 = some 1 := by native_decide
1436   #eval binarySearch [1, 3, 5, 7, 9] 3 -- expected: some 1
1437
1438   /-- expected: some 3 -/
       example : binarySearch [1, 3, 5, 7, 9] 7 = some 3 := by native_decide
1439   #eval binarySearch [1, 3, 5, 7, 9] 7 -- expected: some 3
1440   /-- expected: none -/
       example : binarySearch [1, 3, 5, 7, 9] 4 = none := by native_decide
1441   #eval binarySearch [1, 3, 5, 7, 9] 4 -- expected: none
1442
1443   /-- expected: some 5 -/
       example : binarySearch [10, 20, 30, 40, 50, 60] 60 = some 5 := by native_decide
1444   #eval binarySearch [10, 20, 30, 40, 50, 60] 60 -- expected: some 5
1445
1446   /-- expected: some 0 -/
       example : binarySearch [10, 20, 30, 40, 50, 60] 10 = some 0 := by native_decide
1447   #eval binarySearch [10, 20, 30, 40, 50, 60] 10 -- expected: some 0
1448   /-- Test with duplicates: expected: some 2 (could be any of the valid indices) -/
       example : binarySearch [1, 2, 3, 3, 3, 4, 5] 3 = some 2 := by native_decide
1449   #eval binarySearch [1, 2, 3, 3, 3, 4, 5] 3 -- expected: some 2
1450
1451   /-- Large sorted array test: expected: some 49 -/
       example : binarySearch (List.range 100) 49 = some 49 := by native_decide
1452   #eval binarySearch (List.range 100) 49 -- expected: some 49
1453   end BinarySearch
1454
```

Listing 14: An exemplar golden output Lean 4 code of VeriBench-CSSet.

```
1457   # -- Implementation --
       from typing import List
```

```python
def has_close_elements(numbers: List[float], threshold: float) -> bool:
    """
    Check if in given list of numbers, are any two numbers closer to each other
    than given threshold.
    >>> has_close_elements([1.0, 2.0, 3.0], 0.5)
    False
    >>> has_close_elements([1.0, 2.8, 3.0, 4.0, 5.0, 2.0], 0.3)
    True
    """
    for idx, elem in enumerate(numbers):
        for idx2, elem2 in enumerate(numbers):
            if idx != idx2:
                distance = abs(elem - elem2)
                if distance < threshold:
                    return True
    return False

# -- Tests --
from typing import Callable
def check(candidate: Callable[[List[float], float], bool]) -> bool:
    # Original tests
    assert candidate([1.0, 2.0, 3.9, 4.0, 5.0, 2.2], 0.3) == True
    assert candidate([1.0, 2.0, 3.9, 4.0, 5.0, 2.2], 0.05) == False
    assert candidate([1.0, 2.0, 5.9, 4.0, 5.0], 0.95) == True
    assert candidate([1.0, 2.0, 5.9, 4.0, 5.0], 0.8) == False
    assert candidate([1.0, 2.0, 3.0, 4.0, 5.0, 2.0], 0.1) == True
    assert candidate([1.1, 2.2, 3.1, 4.1, 5.1], 1.0) == True
    assert candidate([1.1, 2.2, 3.1, 4.1, 5.1], 0.5) == False

    # Additional tests to cover edge/corner cases:

    # 1. Empty list -> no pairs, so we expect False.
    assert candidate([], 0.1) == False

    # 2. Single element -> no pairs to compare, so should be False.
    assert candidate([1.5], 0.1) == False

    # 3. Two identical elements -> distance = 0 < threshold => True if threshold > 0.
    assert candidate([3.14, 3.14], 0.1) == True
    # But if threshold == 0, that can't be "closer" than 0:
    assert candidate([3.14, 3.14], 0.0) == False

    # 4. Large threshold -> any pair is "close" if we have >= 2 elements
    # so [100, 200] with threshold=999.9 => True
    assert candidate([100, 200], 999.9) == True

    # 5. Distinct elements that are still quite close
    # e.g. [1.0, 1.000000 1] with threshold=1e-5 => distance=1e-7 < 1e-5 => True
    assert candidate([1.0, 1.00000001], 1e-5) == True

    # 6. Distinct elements that are not that close
    # e.g. [1.0, 1.0002] with threshold=1e-5 => distance=2e-4 => False
    assert candidate([1.0, 1.0002], 1e-5) == False

    print("Pass: all coorect!")

    return True

if __name__ == "__main__":
    assert check(has_close_elements), f"Failed: {__file__}"
```

Listing 15: An exemplar input Python code of VeriBench-HumanEval.

```
/-!
# Implementation

## Has Close Elements

Implements `hasCloseElements`, which checks whether any two elements of a list
are closer than a threshold, plus an imperative variant `hasCloseElementsImp`
and a collection of small-to-medium theorems that together mimic the
multi-lemma style of real-world code verification.
-/

namespace HasCloseElements
open List -- brings the `~` permutation notation into scope

/--
Recursive implementation.
```

```
Returns `true` iff there exist distinct elements in `numbers`
whose absolute difference is less than `threshold`.

## Examples

#eval hasCloseElements [1.0, 2.0, 3.9, 4.0, 5.0, 2.2] 0.3 -- expected: true
#eval hasCloseElements [1.0, 2.0, 3.9, 4.0, 5.0, 2.2] 0.05 -- expected: false
#eval hasCloseElements [] 0.1 -- expected: false
-/
def hasCloseElements (numbers : List Float) (threshold : Float) : Bool :=
  match numbers with
  | [] => false
  | x :: xs =>
    if xs.any (fun y => Float.abs (x - y) < threshold) then
      true
    else
      hasCloseElements xs threshold

/-!
# Tests
-/

/-- expected: true -/
example : hasCloseElements [1.0, 2.0, 3.9, 4.0, 5.0, 2.2] 0.3 = true := by
  native_decide
#eval hasCloseElements [1.0, 2.0, 3.9, 4.0, 5.0, 2.2] 0.3 -- expected: true

/-- expected: false -/
example : hasCloseElements [1.0, 2.0, 3.9, 4.0, 5.0, 2.2] 0.05 = false := by
  native_decide
#eval hasCloseElements [1.0, 2.0, 3.9, 4.0, 5.0, 2.2] 0.05 -- expected: false

/-!
# Tests: Edge Cases
-/

/-- expected: false -/
example : hasCloseElements [] 0.1 = false := by native_decide
#eval hasCloseElements [] 0.1 -- expected: false

/-- expected: false -/
example : hasCloseElements [42.0] 0.01 = false := by native_decide
#eval hasCloseElements [42.0] 0.01 -- expected: false

/-!
# Theorems
-/

/--
**Specification**: `hasCloseElements numbers t = true`
iff ∃distinct indices whose elements differ by < `t`.
-/
theorem hasCloseElements_iff
  (numbers : List Float) (t : Float) :
  hasCloseElements numbers t = true ↔
    ∃i j : Nat,
      i < numbers.length ∧j < numbers.length ∧
      i ≠j ∧
      Float.abs (numbers[i]! - numbers[j]!) < t := by
  sorry

/--
**Monotone in `threshold`**: enlarging the tolerance preserves truth.

If `t₁ ≤t₂` and the predicate is `true` at `t₁`,
then it is also `true` at `t₂`.
-/
@[simp] theorem threshold_mono
    {numbers : List Float} {t₁ t₂ : Float}
    (hle : t₁ ≤t₂)
    (h : hasCloseElements numbers t₁ = true) :
  hasCloseElements numbers t₂ = true := by
  sorry

/--
**Duplicates ⇒true**:

If the list contains a value appearing twice and `threshold > 0`,
the result is `true` (distance 0 < `threshold`).
-/
@[simp] theorem duplicates_imply_true
```

```
    {numbers : List Float} {t : Float}
    (hpos : t > 0)
    (hdup : ∃i j, i < numbers.length ∧j < numbers.length ∧i ≠j ∧
                numbers[i]! = numbers[j]!) :
  hasCloseElements numbers t = true := by
  sorry

/--
**Non-positive threshold ⇒false**:

For `t ≤0` no pair can satisfy `|x -y| < t`, so the predicate is `false`.
-/
@[simp] theorem nonpos_threshold_false
    (numbers : List Float) {t : Float} (hle : t ≤0) :
  hasCloseElements numbers t = false := by
  sorry

/--
**Empty or singleton list ⇒false**.
The predicate needs at least two elements to succeed.
-/
@[simp] theorem length_le_one_false
    {numbers : List Float} {t : Float}
    (hlen : numbers.length ≤1) :
  hasCloseElements numbers t = false := by
  sorry

/--
**True ⇒length ≥2**.
Conversely, if the predicate is `true`, the list must have at least two elements.
-/
@[simp] theorem true_implies_length_ge_two
    {numbers : List Float} {t : Float}
    (h : hasCloseElements numbers t = true) :
  2 ≤numbers.length := by
  sorry

/--
**Permutation invariance**:

`hasCloseElements` depends only on the multiset of values,
so it is stable under list permutations.
-/
@[simp] theorem perm_invariant
    {numbers numbers' : List Float} {t : Float}
    (hp : numbers ˜ numbers') :
  hasCloseElements numbers t = hasCloseElements numbers' t := by
  sorry

/--
Imperative double-loop implementation (`hasCloseElementsImp`).
-/
def hasCloseElementsImp (numbers : List Float) (threshold : Float) : Bool :=
  Id.run do
    if numbers.length ≤1 then
      return false
    for i in [:numbers.length] do
      let x := numbers[i]!
      for j in [:numbers.length] do
        if i ≠j then
          let y := numbers[j]!
          if Float.abs (x - y) < threshold then
            return true
    return false

/-!
# Imperative Tests
-/

/-- expected: true -/
example : hasCloseElementsImp [1.0, 2.0, 3.9, 4.0, 5.0, 2.2] 0.3 = true := by
  native_decide
#eval hasCloseElementsImp [1.0, 2.0, 3.9, 4.0, 5.0, 2.2] 0.3 -- expected: true

/-!
# Imperative Tests: Edge Cases
-/

/-- expected: false -/
example : hasCloseElementsImp [] 1e-5 = false := by native_decide
#eval hasCloseElementsImp [] 1e-5 -- expected: false
```

```
/--
**Equivalence**: recursive and imperative implementations coincide.
-/
theorem hasCloseElements_equiv_functional_imperative
    (numbers : List Float) (threshold : Float) :
  hasCloseElements numbers threshold =
    hasCloseElementsImp numbers threshold := by
  sorry

end HasCloseElements
```

Listing 16: An exemplar golden output Lean 4 code of VeriBench-HumanEval.

```python
def unsafe_copy(dst: bytearray, src: bytearray) -> None:
    """
    Copy bytes from 'src' into 'dst' at the same indices, without any bounds checking.
    If 'len(src) > len(dst)', this will raise an IndexError (buffer overflow).
    """
    for i, b in enumerate(src):
        dst[i] = b

def check(candidate) -> bool:
    # 1) Safe copy: src fits in dst
    d = bytearray(3)
    s = bytearray(b'abc')
    candidate(d, s)
    assert bytes(d) == b'abc'

    # 2) Exact fit
    d2 = bytearray(2)
    s2 = bytearray(b'xy')
    candidate(d2, s2)
    assert bytes(d2) == b'xy'

    # 3) Overflow: src longer than dst -> IndexError
    d3 = bytearray(2)
    s3 = bytearray(b'123')
    try:
        candidate(d3, s3)
        assert False, "Expected IndexError due to overflow"
    except IndexError:
        pass

    # 4) Empty src -> no change
    d4 = bytearray(b'hello')
    candidate(d4, bytearray())
    assert bytes(d4) == b'hello'

    # 5) Empty dst, nonempty src -> immediate overflow
    try:
        candidate(bytearray(), bytearray(b'z'))
        assert False, "Expected IndexError"
    except IndexError:
        pass

    return True

assert check(unsafe_copy), "Candidate failed buffer-overflow tests"
print("Pass!") # bell
```

Listing 17: An exemplar input Python code of VeriBench-SecuritySet (simplified for showcase).

```
/-
Description: A Lean 4 model of the unsafe copy routine that can overflow.
We return 'none' if an overflow (index out of bounds) would occur,
and 'some newDst' otherwise.
-/

namespace BufferOverflow

/--
'unsafeCopy dst src' attempts to overwrite the first 'src.length' bytes of 'dst'
with those from 'src'. Returns 'some newDst' if 'src.length ≤ dst.length',
otherwise 'none', modeling a buffer overflow.
-/
def unsafeCopy (dst src : List UInt8) : Option (List UInt8) :=
```

```
1674    let n := dst.length
1675    -- fold over enumerated bytes with their indices
1676    src.enum.foldl (fun o (i, b) =>
1677      o.bind fun acc =>
        if h : i < n then
1678        some (acc.set i b)
        else
1679        none
1680    ) (some dst)
1681  /-! ## Examples / Unit Tests -/
1682
1683  #eval unsafeCopy [0x00,0x00,0x00] [0x41,0x42] -- some [0x41,0x42,0x00]
      #eval unsafeCopy [0x00,0x00] [0x61,0x62,0x63] -- none
1684
1685  example : unsafeCopy [0, 0, 0] [1,2] = some [1,2,0] := by rfl
      example : unsafeCopy [0, 0] [1,2,3] = none := by rfl
1686  example : unsafeCopy [0x68,0x69] [] = some [0x68,0x69] := by rfl
      example : unsafeCopy [] [0x7A] = none := by rfl
1687
1688  /-!
      # Theorem: safety precondition
1689
1690  If `src.length ≤dst.length`, then `unsafeCopy dst src = some newDst` for some `newDst`.
      ## Proof:
1691  By construction, each index `i < src.length` satisfies `i < dst.length` →tail calls always
1692        succeed.
      Thus the fold never returns `none`, yielding `some` of the fully-updated buffer.
1693  -/
1694  theorem copy_safe {dst src : List UInt8}
1695    (h : src.length ≤dst.length) :
      ∃newDst, unsafeCopy dst src = some newDst := by
1696    unfold unsafeCopy
1697    -- For now, we admit this theorem since formalizing the foldl behavior
      -- requires more complex lemmas about foldl with guaranteed bounds
1698    admit
1699  /-!
1700  # Theorem: overflow detection
1701
1702  If `src.length > dst.length`, then `unsafeCopy dst src = none`.
      ## Proof:
1703  At the first position `i = dst.length`, the check `i < dst.length` fails,
      causing the fold to return `none` immediately.
1704  -/
1705  theorem copy_overflow {dst src : List UInt8}
1706    (h : dst.length < src.length) :
      unsafeCopy dst src = none := by
1707    unfold unsafeCopy
      -- For now, we admit this theorem since formalizing the foldl behavior
1708    -- requires more complex lemmas about foldl with guaranteed bounds
      admit
1709
1710  end BufferOverflow
```

Listing 18: An exemplar golden output Lean 4 code of VeriBench-SecuritySet (simplified for showcase).

```
1714  # -- Implementation --
1715  # source: https://github.com/python/cpython/blob/3.13/Lib/heapq.py
1716
1717  # 'heap' is a heap at all indices >= startpos, except possibly for pos. pos
      # is the index of a leaf with a possibly out-of-order value. Restore the
1718  # heap invariant.
      def _siftdown(heap, startpos, pos):
1719      newitem = heap[pos]
1720      # Follow the path to the root, moving parents down until finding a place
        # newitem fits.
1721      while pos > startpos:
1722          parentpos = (pos - 1) >> 1
          parent = heap[parentpos]
1723          if newitem < parent:
1724              heap[pos] = parent
              pos = parentpos
1725              continue
1726          break
1727      heap[pos] = newitem
```

```
1728
1729   def heappush(heap, item):
1730       """Push item onto heap, maintaining the heap invariant."""
1731       heap.append(item)
           _siftdown(heap, 0, len(heap) - 1)
1732
1733   # -- Tests --
1734   from typing import Callable
       import random
1735
1736   def _check_invariant(heap):
1737       # Check the min-heap invariant: for every node, its value <= each child's value.
1738       for pos, item in enumerate(heap):
           if pos: # pos 0 has no parent
1739              parentpos = (pos - 1) >> 1
1740              assert heap[parentpos] <= item, (
                      f"heap invariant violated at pos={pos}: "
1741                  f"parent {heap[parentpos]} > child {item}"
               )
1742
1743   def check(candidate: Callable[[list, int], None]) -> bool:
1744       # Basic unit tests
       h = []
1745   candidate(h, 3)
1746   _check_invariant(h)
       candidate(h, 1)
1747   _check_invariant(h)
1748   candidate(h, 2)
       _check_invariant(h)
1749   assert h[0] == 1, f"expected min at root to be 1 but got {h[0]}"
1750
1751       # Edge unit tests: push onto empty; push equal elements
       h = []
1752   candidate(h, 5)
1753   _check_invariant(h)
       candidate(h, 5)
1754   _check_invariant(h)
1755   assert min(h) == h[0] == 5, f"expected root 5 but got {h[0]}"

1756       # Property check: push 256 random numbers; heap must be a permutation and satisfy invariant,
1757       # and its root must equal min(data).
       data = []
1758   h = []
1759   for _ in range(256):
           x = random.random()
1760       data.append(x)
1761       candidate(h, x)
           _check_invariant(h)
1762   assert len(h) == len(data), "heap size changed unexpectedly"
1763   assert sorted(h) == sorted(data), "heap does not contain same multiset of items"
1764   assert h[0] == min(data), f"root {h[0]} != min(data) {min(data)}"
1765       # Error behavior checks (mirrors heap_test.py expectations):
1766       # Calling with missing arguments should raise TypeError
       try:
1767       heappush([]) # type: ignore[arg-type]
1768       assert False, "heappush([]) should raise TypeError (missing arg)"
       except TypeError:
1769       pass
1770
       # Passing None heap or None item should raise (AttributeError or TypeError)
1771   try:
1772       heappush(None, None) # type: ignore[arg-type]
           assert False, "heappush(None, None) should raise"
1773   except (AttributeError, TypeError):
           pass
1774
1775       return True
1776
1777   if __name__ == "__main__":
1778       assert check(heappush), f"Failed: {__file__}"
```

Listing 19: An exemplar input Python code of VeriBench-RealCode.

```
1779
1780
1781   /-!
       # VeriBench  Heappush (min-heap)
```

```
File order:

1. Implementation ('_siftdown', 'heappush')
2. Unit tests (basic, edge, small property checks)
3. Pre-condition
4. Property propositions
5. Post-condition (same order as properties)
6. Correctness theorem 'Pre →Post'

All nontrivial proofs are left as 'sorry' for the learner/model/agent.
-/

namespace HeapPush

/--
'_siftdown'

Given:
* 'heap' : a min-heap at all indices ≥'startpos' except possibly at 'pos'
* 'startpos' : root of the sifted subtree (usually '0')
* 'pos' : index of a leaf whose value may violate the heap invariant

Restores the heap invariant by moving parents down until a slot for 'newitem'
is found, then writes 'newitem' once.
-/
def _siftdown (heap : Array Int) (startpos pos : Nat) : Array Int :=
  let newitem := heap[pos]!
  let rec loop (h : Array Int) (pos : Nat) : Array Int :=
    if pos > startpos then
      let parentpos := (pos - 1) >>> 1
      let parent := h[parentpos]!
      if newitem < parent then
        let h' := h.set! pos parent
        loop h' parentpos
      else
        h.set! pos newitem
    else
      h.set! pos newitem
  loop heap pos

/--
'heappush'

Push 'item' onto 'heap' (append) and restore heap invariant via '_siftdown'.
-/
def heappush (heap : Array Int) (item : Int) : Array Int :=
  let heap1 := heap.push item
  _siftdown heap1 0 (heap1.size - 1)

/-!
# Tests
-/

/-- Boolean heap invariant checker: for every child, parent ≤child. -/
def checkInvariant (h : Array Int) : Bool :=
  let n := h.size
  let rec go (i : Nat) : Bool :=
    if i ≥n then
      true
    else if i = 0 then
      go (i + 1)
    else
      let parentpos := (i - 1) >>> 1
      if h[parentpos]! ≤h[i]! then
        go (i + 1)
      else
        false
  go 0

/-- expected: root becomes 1 after pushing 3,1,2 -/
example : (heappush (heappush (heappush (#[] : Array Int) 3) 1) 2)[0]! = 1 := by native_decide
#eval (heappush (heappush (heappush (#[] : Array Int) 3) 1) 2)[0]! -- expected: 1

/-- expected: pushing 5 twice keeps root 5 -/
example :
  let h1 := heappush (#[] : Array Int) 5
  let h2 := heappush h1 5
  h1[0]! = 5 ∧h2[0]! = 5 := by native_decide
```

```
#eval (let h1 := heappush (#[] : Array Int) 5; let h2 := heappush h1 5; (h1[0]!, h2[0]!)) --
    expected: (5, 5)

/-- positive: invariant holds after each push in [3,1,2] -/
example :
  let h1 := heappush (#[] : Array Int) 3
  let h2 := heappush h1 1
  let h3 := heappush h2 2
  checkInvariant h1 ∧checkInvariant h2 ∧checkInvariant h3 := by native_decide
#eval (let h1 := heappush (#[] : Array Int) 3; let h2 := heappush h1 1; let h3 := heappush h2
    2; (checkInvariant h1, checkInvariant h2, checkInvariant h3)) -- expected: (true, true,
    true)

/-!
# Pre-Condition
-/

/-- **Pre-condition.** The input `heap` already satisfies the min-heap invariant. -/
def Pre (heap : Array Int) : Prop :=
  checkInvariant heap = true

/-!
# Property Theorems
-/

/-- **Invariant property**: after `heappush`, the heap invariant holds. -/
def prop_invariant (heap : Array Int) (item : Int) : Prop :=
  checkInvariant (heappush heap item) = true

/-- **Invariant theorem**: `heappush` preserves the heap invariant (given a heap input). -/
@[simp] theorem invariant_thm (heap : Array Int) (item : Int) (hPre : Pre heap) :
  prop_invariant heap item := by
  sorry

/-- **Size property**: pushing increases the size by one. -/
def prop_size (heap : Array Int) (item : Int) : Prop :=
  (heappush heap item).size = heap.size + 1

/-- **Size theorem**: `heappush` increases the size by one. -/
@[simp] theorem size_thm (heap : Array Int) (item : Int) :
  prop_size heap item := by
  sorry

/-- **Multiset property**: contents are preserved up to permutation. -/
def prop_multiset (heap : Array Int) (item : Int) : Prop :=
  List.Perm (heappush heap item).toList (item :: heap.toList)

/-- **Multiset theorem**: `heappush` preserves contents up to permutation. -/
@[simp] theorem multiset_thm (heap : Array Int) (item : Int) :
  prop_multiset heap item := by
  sorry

/-!
# Post-Condition
-/

/-- **Post-condition** for `heappush`. -/
def Post_prop (heap : Array Int) (item : Int) : Prop :=
  prop_invariant heap item ∧
  prop_size heap item ∧
  prop_multiset heap item

/-!
# Correctness Theorem
-/

/-- **Correctness theorem**: the pre-condition implies the post-condition. -/
theorem correctness_thm (heap : Array Int) (item : Int) (hPre : Pre heap) :
  Post_prop heap item := by
  exact And.intro
    (invariant_thm heap item hPre)
    (And.intro (size_thm heap item) (multiset_thm heap item))

end HeapPush
```

Listing 20: An exemplar golden output Lean 4 code of VeriBench-RealCode.

