# OpenReview forum: "VeriBench: End-to-End Formal Verification Benchmark for AI Code Generation in Lean 4"
_ICLR.cc/2026/Conference — Submitted to ICLR 2026_

### Official Review · Reviewer_Q2Er · 2025-10-19

**Soundness:** 2
**Presentation:** 3
**Contribution:** 2
**Rating:** 2
**Confidence:** 4

**Summary:**

The paper introduces VeriBench, a benchmark for evaluating large language models’ (LLMs) ability to generate provably correct programs in Lean 4.
VeriBench requires models to translate Python code into Lean 4 programs that include implementations, unit tests, specifications/theorems, and machine-checked proofs, offering an end-to-end assessment of formal code verification capabilities.
The benchmark spans 140 tasks across five difficulty levels — HumanEval, EasySet, CSSet, SecuritySet, and RealCodeSet — combining classical algorithmic and real-world security-critical programs.
Results show that current models (e.g., Claude 3.7 Sonnet, DeepSeek-Prover V2) struggle severely with formal proofs and real-code verification, highlighting proof synthesis as a key bottleneck. VeriBench thus establishes a rigorous foundation for advancing trustworthy AI-assisted formal verification.

**Strengths:**

- End-to-End Benchmark

VeriBench uniquely integrates all stages of formal code generation, implementation, testing, specification, and proof within a single Lean 4 environment.

- Diversity

Inclusion of security-critical and production-grade programs (from MIT 6.858 and Python stdlib) moves beyond synthetic exercises.

- Evaluation Framework

Introduce Trace-based feedback-driven agents

**Weaknesses:**

- Limited Scope and Generality.


- Limited Scale and Coverage

Despite diverse subsets, all benchmarks are Python-to-Lean 4 translations in a limited number.
Its total of 140 tasks remains small compared to other large-scale code or theorem-proving datasets.
This limited scale constrains statistical robustness and may not fully reflect model generalization across domains, paradigms, or theorem types. In particular, the RealCodeSet and SecuritySet are valuable but too small (only 5 and 28 programs respectively) to support strong empirical conclusions. Besides, the benchmark assumes that LLMs can semantically align Python code, Lean syntax, and proof structures, which in practice remains difficult.

- Reliance on the LLM Judge for the theorem evaluation

The dataset’s generation pipeline partially relies on LLM-assisted curation (o3, Claude) before human review. While human validation mitigates bias, it introduces a risk of training contamination — future or current LLMs may have seen similar Python or HumanEval code during pretraining.
Moreover, because the Lean translations are bootstrapped from AI models, they may implicitly encode stylistic or syntactic priors that align with specific model outputs, reducing benchmark neutrality.

**Questions:**

- Given that theorem sets cannot be exhaustive, how do the authors ensure that omitted properties do not unfairly penalize or reward certain model behaviors? Could a probabilistic coverage measure or human validation subset be added?

- The paper validates the LLM judge’s monotonicity and consistency, but how robust is it across architectures (e.g., GPT-4 vs. Claude)? Would ensemble or symbolic judges improve objectivity?

- Since the paper mentions plans to extend VeriBench to multi-language settings, how do the authors envision handling semantic gaps (e.g., between imperative C++ and functional Lean 4) while maintaining proof-level correspondence?

---

### Official Review · Reviewer_2bBm · 2025-10-30

**Soundness:** 2
**Presentation:** 1
**Contribution:** 2
**Rating:** 2
**Confidence:** 4

**Summary:**

This paper introduces VeriBench, a benchmark for evaluating LLMs' capabilities in end-to-end formal code verification using Lean 4. The benchmark comprises 140 tasks across five difficulty levels (HumanEval, EasySet, CSSet, SecuritySet, RealCodeSet), requiring models to translate Python code into Lean 4 with complete implementations, tests, specifications, and machine-checked proofs. The authors propose four evaluation subtasks and introduce an agentic evaluation framework using Trace, along with a validated LLM judge for theorem quality assessment.

**Strengths:**

- Comprehensive benchmark design: The inclusion of security-critical programs (MIT 6.858 labs) and real production code (Python standard library) is a significant advancement over existing benchmarks that focus primarily on textbook algorithms.
- Agentic framework: The Trace-based evaluation with self-debug and self-improve variants demonstrates the value of feedback-driven approaches

**Weaknesses:**

- Poor Presentation: The writing is difficult to follow, and the formatting in some paragraphs is broken (e.g., Section 5.1 LLM Judge and Lines 852-855).
- Incomplete End-to-End Evaluation: The evaluation lacks a measurement of proof verification success, such as whether generated theorems are actually proven, rather than just passing tests or receiving LLM-generated scores.
- Limited Experimental Analysis: There is no comparison of model-generated theorems against gold theorems to identify semantic gaps. Additionally, the discussion of why self-improvement performs worse than self-debugging is limited.
- Questionable Benchmark Task Quality: For the theorems in Table 1, it's unclear if they can truly be proven and have ground-truth proofs, as many "sorry" and "admit" statements were found in the gold examples.

**Questions:**

- Regarding Theorem Proving:
  - What percentage of the "gold standard" theorems in Table 1 have complete, verifiable proofs, as opposed to those marked "sorry"?
  - Can these theorems truly be proven with ground-truth proofs?
- Regarding Model Performance:
  - Please provide a failure case analysis, including specific examples where all models fail and the reasons why.
  - For end-to-end generation, what percentage of model-generated proofs successfully verify without needing "sorry"?
  - Why does the "self-improve" method underperform "self-debug"?
- Regarding Evaluation:
  - What is the quality of the test cases used to evaluate the specifications and code?

**Details Of Ethics Concerns:**

This paper introduces VeriBench, a benchmark for evaluating LLMs' capabilities in end-to-end formal code verification using Lean 4. The benchmark comprises 140 tasks across five difficulty levels (HumanEval, EasySet, CSSet, SecuritySet, RealCodeSet), requiring models to translate Python code into Lean 4 with complete implementations, tests, specifications, and machine-checked proofs. The authors propose four evaluation subtasks and introduce an agentic evaluation framework using Trace, along with a validated LLM judge for theorem quality assessment.

---

### Official Review · Reviewer_K6Sq · 2025-11-03

**Soundness:** 2
**Presentation:** 1
**Contribution:** 3
**Rating:** 6
**Confidence:** 4

**Summary:**

This paper presents VERIBENCH, a comprehensive benchmark aimed at evaluating end-to-end formal verification in AI code generation using Lean 4. VERIBENCH’s tasks require large language models to translate reference Python code into fully verified Lean 4 programs, including implementations, unit tests, specifications, and machine-checked proofs. The benchmark comprises 140 tasks spanning five difficulty levels, with tasks sourced from HumanEval, foundational CS exercises, standard library code, classical algorithms, and adapted real-world security-critical programs. The paper introduces agentic evaluation architectures and proposes a hierarchical metric suite, including compilation, test, theorem, and proof success rates. Evaluation across LLMs/theorem provers and agentic strategies uncovers substantial limitations of current LLMs, highlights feedback-driven improvements, and details a methodology for certifying the trustworthiness of LLM-based evaluation judges.

**Strengths:**

1. Comprehensive, End-to-End Scope: Each benchmark item includes Python code, Lean translation, unit tests, formal theorem statements, and machine-checked proofs, allowing for holistic, multi-stage evaluation
2. Breadth of Benchmark: VERIBENCH spans a uniquely rich spectrum of tasks, incorporating both textbook-style problems and authentic security-critical code drawn from real-world vulnerabilities
3. Contribution to Methodology: Describes a principled approach for verifying LLM judge trustworthiness, with sanity checks for monotonicity, correctness, and completeness, also the metrics for evaluation are nice

**Weaknesses:**

1. Uneven distribution of various problem types: as the author specified at line 166, this benchmark is essentially assembled from 4 subsets, yet the size of each subset differs. The overall scope of this benchmark is also limited, which pose the question as to whether this small size would be representative enough of the complexity of this task.

2. small set of tested models: The set of models evaluated in this paper is quite limited, I feel that it warrants to put more SOTA models and if possible larger theorem provers e.g. DeepSeekProver671B, GoedelProver etc.

3. Related work: is currently a big chunk and should visualize this comparison in a table to make it easier (e.g. 3 column: Name+Content+Diff with VeriBench). The current related work paragraphs seem (at least partially) LM-generated with its excessive use of em-dash. It would be better if this related work section brings more work that compares Lean+Computer Security/Theoretical Computer Science related work into comparison (e.g. in AI4MATH workshops)

While I can see there's related work in appendix, still one should present the related work section better, because currently it's very hard to read and grasp the key idea there as it's too chunky and lacks a coherent central line of story.

**Questions:**

Many nuances where the manuscript could be improved:
Line 270 onwards: many repetitive appendix K in each para, could be merge into one umbrella signpost to Appendix K and instead maybe make a figure to show each subset, how it's curated and the capability it's supposed to evaluate (e.g. I can imagine a Venn diagram of how each subset tests different sets of capability would be nice to see)

Spacing issue: e.g. Figure 1 takes a lot of space that could have been reallocated for 2 figures (this seems like the authors recently converted from 2-column format to single-column) which I understand is tricky, nevertheless this warrants further improvement for better presentation

Overall I think the ideas in this work are generally nice, but the presentation could be massively improved as the current form strikes me as highly coarse and maybe (speculatively) the product of a rushing timeline, so it would be great if the authors could further polish the presentation of this work, the scope and size of both number of questions and number of models covered in evaluation also makes me believe that more time put into this work may present a much better contribution to the field, hence my rating.

---

### Official Review · Reviewer_mnVJ · 2025-11-05

**Soundness:** 2
**Presentation:** 2
**Contribution:** 2
**Rating:** 2
**Confidence:** 4

**Summary:**

This paper introduces VeriBench, a benchmark designed to evaluate large language models (LLMs) on end-to-end formal verification tasks. VeriBench aims to measure a model’s ability to generate provably correct Lean 4 code from Python references, including implementations, tests, specifications, theorems, and proofs. The benchmark contains 140 tasks divided into five subsets: HumanEval, EasySet, CSSet, SecuritySet, and RealCodeSet. The authors implement a feedback-driven evaluation pipeline using TRACE and DSPy agents and propose an LLM-based judge to assess theorem quality. Experiments show that current LLMs struggle with theorem proving and compilation, while iterative, self-debugging agents achieve moderate improvements.

**Strengths:**

1. Tackles a timely and critical topic: formal verification of AI-generated code.


2. Diverse benchmark composition spanning five subsets, including real-world and security-critical programs.


3. Inclusion of agentic, feedback-driven evaluation (TRACE, DSPy) reflecting current trends in LLM reasoning.


4. Comprehensive baseline coverage across multiple model families.


5. The trust-validation sanity checks for the LLM judge, though limited, show some attempt at methodological care.

**Weaknesses:**

1. Unclear Benchmark Definition
    - The task definition is inconsistent. The abstract claims “complete program implementations with tests, theorems, and proofs,” but Section 3 states “translate Python code with docstring and unit tests to Lean 4 implementation.” It is unclear whether unit tests are an input or output.


    - The precise benchmark tasks and their expected inputs/outputs are not formally specified.


    - In HumanEval, only 56 of 164 tasks are included, but selection criteria are unexplained.


    - The role of the imperative version in Lean 4 is ambiguous. Lean 4 is fundamentally functional; if imperative variants use monads, this should be clarified.


    - In the SecuritySet, the translation of vulnerabilities from Python to Lean 4 is unclear. Are tasks meant to repair vulnerabilities or merely prove properties?


    - The dataset’s content boundaries are vague. Does it include correctness/equivalence proofs or only expect them to be generated?


    - No transparent process is given for validating theorems or ensuring dataset quality beyond “human + AI curation.”


2. Methodological and Evaluation Issues
    - Example “golden” outputs (Listing 2) include sorry placeholders, proof stubs that invalidate verification integrity. The paper should state whether such cheats are filtered.


    - The proof success metric is insufficiently defined: does one theorem failure mark the whole program as failed, or are partial passes counted?


    - Reliance on an LLM judge for correctness evaluation is not rigorous. Although the authors test monotonicity and consistency, such heuristics do not guarantee semantic faithfulness.


    - Using the same model (Claude 3.7) as both agent and judge risks self-bias and metric circularity.


3. Lack of Analysis and Insight
    - Results tables mostly show that models perform poorly (e.g., 0 % theorem-proving success on RealCodeSet) but offer no qualitative or failure-mode analysis.


    - The claim that self-debug agents improve compilation from 35 %→49 % lacks breakdown of error categories or examples of corrected failure modes.


    - The paper thus provides little scientific insight beyond “the task is hard.”


4. Presentation and Technical Clarity
    - Several notational and typographical errors impede understanding:
        - Section 4: inconsistent Post(x) vs. Post(x, Prog) definitions.
        - Line 193: “Input →” missing predicate; Line 197: “; (x) ==>” likely meant Pre(x).


    - Minor errors:
        - Line 166, “VeriBench consists of four subsets”, should be five subsets;
        - Line 274 and 280, missing trailing dot and incomplete sentence;
        - Line 292, “generation pipeline were a second human” should be “generation pipeline where a second human”;
        - Line 411, duplicate “end-to-end”;
        - Incomplete sentence in “Future work” subsection in Section 9, consider rephrasing them;


5. Incremental Novelty
    - VeriBench largely repackages existing ideas from VERINA (Lean end-to-end tasks), CLEVER (spec + proof), and FVAPPS (machine-generated verification problems).


    - The addition of small real-code and security subsets and agentic evaluation loops does not constitute a substantial methodological advance.

**Questions:**

Please address my comments in the "Weaknesses" section

---

### Official Review · Reviewer_FCFR · 2025-11-06

**Soundness:** 1
**Presentation:** 3
**Contribution:** 2
**Rating:** 2
**Confidence:** 4

**Summary:**

This paper introduces VeriBench, a benchmark to formally verify automated code generation in Python. The paper considers previous benchmarks in the literature and constructs its own benchmark to address some of the shortcomings in the previous benchmarks. It further uses an automated LLM Judge to evaluate the correctness and comprehensiveness of the verification system. Based on that, it defines a new Quality Score. Finally, the paper evaluates the abilities of several LLMs on VeriBench based on its quality score. It reports that existing LLMs struggle on VeriBench while providing automated feedback to the LLMs help them improve their output and achieve higher accuracies.

**Strengths:**

The topic is novel and interesting in my view and the benchmark has the potential to be useful for the community.

The choice of problems in the benchmark are an improvement over the previous benchmarks.

**Weaknesses:**

- My main concern is that results are mostly based on an LLM judge that may not be reliable and the paper does not make a convincing effort that LLM judge is reliable. My second main concern is that the quality score is based on comprehensiveness which seems subjective and not a concrete measure that can be reliably evaluated and generalized. The paper states: "comprehensive is difficult if not impossible to guarantee even during the gold reference generation of the benchmark."

- I would like to see clear justifications about the quality score and possibly other scores that are less subjective. It would be best if the quality score is somehow compared with other measures such as human judgement.

- I find the method used by the paper for the LLM judge premature and unconvincing. Specifically, the paper does not establish that the LLM judge is reliable enough for a scientific publication. We have results in the literature that have reported very high accuracies using LLM judges while subsequent work have demonstrated that those LLM judges were not reliable overall. For example, Herald translator reported accuracy of 94% on miniF2F evaluated by an LLM judge -- they also evaluated their judge on some small subset suggesting that it is reliable. Later, it was revealed that accuracy of 94% was inaccurate and the correct accuracy was 67%. I would like to see concrete and reliable evaluation of results by this paper. For example, if the paper reports that all the evaluations in the paper are human checked, my concern will be resolved.



- I would have liked to see one case study from beginning to end in the appendix as it appears in the dataset with the gold references along with some LLM outputs and the resulting scores.


- Writing can be improved. Sometimes, writing is not smooth. Moreover, in many instances, the methodology is not described clear enough in my view.


- Clearly, many LLMs are not very capable of generating Lean proofs for correctness of generated codes. This, most likely, has to do with the training set of these LLMs. Lean proofs for the correctness of Python codes is not abundantly available and most of such data is new. The related work section in the paper reflects this. It is very likely that these LLMs are not trained on such data while they are trained on large amounts of Python code. It would be insightful if the paper attempts to train a LLM and see if it can gain the capability for proving the correctness of generated codes.

- The efforts of the paper in providing automated feedback is insightful but does not go far enough, in my view. For example, see the work by Goedel Prover v2 which trains its LLM to use the automated feedback generated by the Lean compiler. There are also other work in the literature that use the feedback from Lean compiler programmatically.





-----------------


Minor comments:

- It seems to me that the list of contributions are inflated. Going from one item to another at the end of page 2, in my view, contributions are being repeated in different ways under different bullet points.

- Writing can be improved. Some sentences do not read well, e.g., "We discover model cannot prove a single theorem in this set." It is not clear which model the sentence refers to.

- It is also not so clear to me what the paper means when t says: "whereas starting from a fresh natural-language description would add an unnecessary detour that today’s LLMs no longer require." Does the paper mean that one can directly start from Python code because translating natural language description to Python is trivial for the LLMs? If that's what the paper is trying to say, I don't think this is a correct statement in general. Such sentence should be defined specifically for certain problems and not remain vague for the reader.

**Questions:**

Please see the weaknesses. Any clarifications may be helpful in evaluating the contribution. I'd be happy to revise my score based on the rebuttal.

---

### Meta-Review · Area_Chair_YaWd · 2026-01-07

**Summary:**

- **FCFR** (2): LLM judge reliability concerns, citing prior failures; subjective quality scores lacking human validation
- **mnVJ** (2): Inconsistent benchmark definition (unclear I/O specifications); gold files contain "sorry" placeholders
- **K6Sq** (6): Uneven subset distribution; limited model evaluation; presentation issues
- **2bBm** (2): Incomplete evaluation; no proof verification metrics; limited failure analysis
- **Q2Er** (2): Limited scope (140 tasks); potential data contamination from LLM-assisted curation

**Reviewer Concerns:**

Not applicable since no author responses are made.

**Reviewer Scores:**

Not applicable since no author responses are made.

---

### Decision · Program_Chairs · 2026-01-26

Reject